# Precision Collaboration for Federated Learning

## Abstract

Inherent heterogeneity of local data distributions, which causes inefficient model learning and significant degradation of model performance, has been a key challenge in Federated Learning (FL). So far, plenty of efforts have focused on addressing data heterogeneity by relying on a hypothetical clustering structure or a consistent information sharing mechanism. However, because of the diversity of the real-world local data, these assumptions may be largely violated. In this work, we argue that information sharing is mostly fragmented in the federated network in reality. More specifically, the distribution overlaps are not consistent but scattered among local clients. We propose the concept "Precision Collaboration" which refers to learning from the informative overlaps precisely while avoiding the potential negative transfer induced by others. In particular, we propose to infer the local data manifolds and estimate the exact local data density simultaneously. The learned manifold aims to precisely identify the overlaps from other clients, and the estimated likelihood allows to generate samples from the manifold in an optimal sampling density. Experiments show that our proposed `PCFL` significantly overcomes baselines on benchmarks and a real-world clinical scenario.

## 1 Introduction

Federated learning (FL) has drawn considerable interest from a variety of disciplines in recent years. FL enables collaborative model learning without the need to access the raw data across different clients, which facilitates real-world scenarios where privacy preservation is crucial, such as finance (Yang et al., 2019), healthcare (Xu et al., 2021) and criminal justice (Berk, 2012). While it is common that the data samples in local clients are non-i.i.d., existing research reveals that **data heterogeneity** could lead to non-guaranteed convergence, inconsistent performance and catastrophic forgetting across different clients (Qu et al., 2022). Despite the promise of FL, an increasing concern is how to effectively handle data heterogeneity before FL is applied in real-world data scenarios.

In view of this challenge, an important direction is personalization. A variety of efforts have been made to explore this direction. For example, Ghosh et al. (2020) proposed to cluster the clients according to their sample distributions and build a customized model for each cluster. However, their hypothesis excludes the possibility of knowledge transfer across clusters. Li et al. (2021b) enhanced personalized model learning by introducing a global regularization term, which assumed that the shared knowledge was consistent across all clients.

Considering the diversity of local data, in this paper, we study a more flexible and general scenario where the distribution overlaps could be fragmented as shown in Figure 1 (a). Since the informative and ambiguous data shards exist simultaneously in another client, collaborating with all data could do harm to the model learning. An interesting and challenging problem is how to selectively collaborate with the favorable part of other clients in a privacy-preserving way.

In this paper, we put forward the concept "Precision Collaboration" for fragmented information sharing. To begin with, we argue that data heterogeneity comes from inconsistent local data manifolds. In particular, the data manifolds of different local clients could share different overlaps. Maximizing the benefit of collaboration requires a precise utilization of these overlaps. Moreover, local data are usually gathered from the manifold based on a particular density. If we want to generate data from the manifold, a precise distribution density approximation for each client could facilitate model learning.

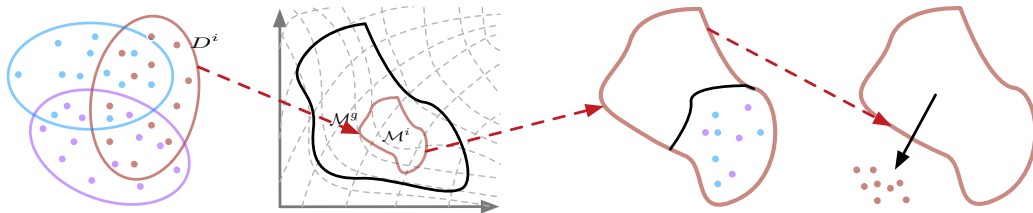

(a) data in local clients   (b) global and local manifold   (c) identify overlapped data   (d) sample from $\mathcal{M}^i$

Figure 1: Overview of our proposed PCFL. (a) Fragmented distribution overlaps exist among clients; (b) learn the global data manifold and determine the local manifold for each client; (c) the data from other clients lie on the local manifold $\mathcal{M}^i$ are identified as informative overlaps; (d) learn a precise local density for synthetic data generation.

To realize our proposed precision collaboration, we develop a novel framework named PCFL shown in Figure 1. We assert that the key to precisely collaborative model learning is identifying and utilizing the distribution overlaps scattered in other clients. These overlaps between clients indeed correspond to a specific data manifold region. We propose to infer the local data manifold to identify the overlaps. While it is hard to learn the local manifold from the insufficient data in local clients directly, we firstly infer the underlying manifold $\mathcal{M}^g$ of the data from all clients, so that the data from all overlapped distributions are utilized for the manifold inference. Then the local manifold $\mathcal{M}^i \subset \mathcal{M}^g$ of the $i$-th client could be determined by local data $D^i$ as shown in Figure 1 (b).

From Figure 1 (c), the local data manifold $\mathcal{M}^i$ is used to identify the beneficial overlaps from other clients. In particular, if a subset of the data from $D^j$ lies on $\mathcal{M}^i$, this subset is the overlaps between the $i$-th and $j$-th clients. To further boost the local model training, we suggest sampling from $\mathcal{M}^i$ with an optimal sampling probability estimated from local data as shown in Figure 1 (d), which effectively mitigates the potential distribution discrepancy. We highlight our key contributions as follows:

- While existing research studies FL under certain assumptions about the information sharing, we investigate a more general learning scenario where the data sharing a common distribution is fragmented among local clients;

- We achieve a more precise collaboration for the federated network by proposing a framework PCFL. Our framework identifies the meaningful overlaps and excludes ambiguous information from other clients, which avoids potential negative transfer;

- PCFL could be used to improve other SOTA algorithms in a plug-and-play way. Empirical experiments corroborate that PCFL significantly outperforms all baselines on a series of benchmark data sets and a real-world clinical data set.

## 2 RELATED WORK

### 2.1 FEDERATED LEARNING AND DATA HETEROGENEITY

Recent years have witnessed growing attention to federated learning (McMahan et al., 2017), of which several challenges have been concerning topics including communication efficiency (Konečný et al., 2016), privacy (Agarwal et al., 2018) and data heterogeneity (Karimireddy et al., 2020). While data heterogeneity could cause the lack of convergence and the potential of catastrophic forgetting (Qu et al., 2022), there are researchers aiming to tackle the heterogeneity by learning a global model. For example, Li et al. (2020) propose a proximal term to restrict the local updates to be closer to the initial model. Mohri et al. (2019) seek a fair model performance distribution by maximizing the model performance on any arbitrary target distribution. Li et al. (2021a) develop MOON that corrects local training by maximizing the agreements of representation between local and global models. Instead of pursuing a balanced performance distribution, we are interested in achieving the best performance for each client by precisely learning the shared informative overlaps from others.

### 2.2 PERSONALIZED FEDERATED LEARNING

In addition to reaching a global consensus, personalized model learning also attracts widespread concern in FL community, which may boost the flexibility of learned models when adapting to local distributions (Cui et al., 2022; Li et al., 2021b). Plentiful research have proposed techniques for a

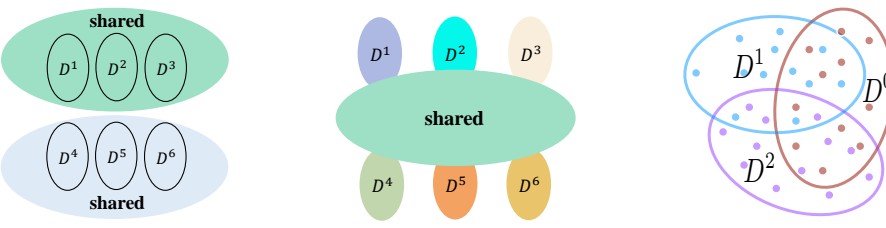

(a) clustered sharing      (b) common partial sharing      (c) fragmented sharing

Figure 2: Illustrations of the three assumptions on data heterogeneity. (a) **clustered sharing**; 1. all information are shared within the clusters; 2. the sharing are not consistent across all clients; (b) **common partial sharing**; 1. partial information within clients are shared; 2. the sharing are consistent across all clients; (c) **fragmented sharing**; 1. partial information within clients may be shared with others, 2. the sharing are not consistent.

trade-off between local and global models. For example, Fallah et al. (2020) proposed to train local models that can quickly adapt to local data starting from an initial shared model in a meta-learning way. Some works train personalized models by interpolating between global and local models (Deng et al., 2020; Dinh et al., 2020). Li et al. (2021b) achieve such a trade-off through regularizing local models close to the global model. There are other works suggesting a partially shared model structure for efficient information transferring (Liang et al., 2020; Collins et al., 2021). Nonetheless, we are concerned that a global model is hard to model the various shared information between clients. The fragmented knowledge requires precise identification when collaboratively learning from others.

To mitigate the potential overfitting when learning from limited local data, some works also attempt generative methods to improve the model performance (Du & Wu, 2020; Zhu et al., 2021). In particular, Zhu et al. (2021) regulate local training with the distilled knowledge from all clients. Du & Wu (2020) lead into GAN for generating similar data for local clients. However, generating data in an arbitrary density could result in distribution discrepancy. An optimal sampling density may present more benefits for local learning tasks.

## 3 NOTATIONS AND PROBLEM DEFINITION

### 3.1 NOTATIONS

Suppose there are $N$ clients in a federated network, each client owns a private dataset $D^k$ with $n^k$ data samples. The dataset $D^k = \{X^k, Y^k\}$ consists of the input space $X^k$ and output space $Y^k$. We use $z = \{x, y\}$ to denote a data point, and $z \in \mathcal{M}$ denotes the data manifold. The input space and the output space are shared across all clients. In the following, we also use $D^i$ to denote the $i$-th client without causing further confusion.

The goal of each client is to learn a best model to predict the label $y$ by collaborating with others. For example, McMahan et al. (2017) propose FedAvg, which learns a global model $f$ for all clients by minimizing the empirical risk over the samples from all clients, i.e.,

$$\min_{f \in \mathcal{F}} \ \frac{1}{\sum_{k=1}^{N} n^k} \sum_{k=1}^{N} \sum_{i=1}^{n^k} l\left(f\left(x_i^k\right), y_i^k\right), \tag{1}$$

where $\mathcal{F}$ is the hypothesis space and $l$ denotes the loss objective of all clients. From Eq.1, FedAvg assumes that the data from different clients associate with a common data manifold $\mathcal{M}$ and sampling density $p_z^g(z)$, i.e.,

$$\forall \, D^i \in \left\{D^0, D^i, ..., D^{N-1}\right\}, \ s.t., \ z \in D^i \subset \mathcal{M}, \ z \sim p_z^g(z). \tag{2}$$

### 3.2 ASSUMPTIONS ON DATA HETEROGENEITY

However, i.i.d. assumption in Eq.(2) is largely violated as the local data distributions may be significantly distinctive. In this event, learning a consensus by averaging the local gradients could cause severe performance degradation on certain clients (Li et al., 2019b; Cui et al., 2021). There are research studying federated learning with non-i.i.d. data and the assumptions on data heterogeneity are mainly from two perspectives.

**Clustered sharing.** As shown Figure 2 (a), the clients partitioned in each cluster own a common data manifolds ($\mathcal{M}^j$) and sampling density ($p_z^j(z)$), i.e.,

$$\forall\, i \in \{0, 1, ..., N-1\}, \exists\, j \in \{0, 1, ..., K\}\, (K < N),\ s.t.,\ z \in D^i \subset \mathcal{M}^j\ and\ z \sim p_z^j(z). \quad (3)$$

From Eq.(3), clustered sharing requires that all message is shared within the clusters, and there is no knowledge transferring across clusters.

**Common partial sharing.** From Figure 2 (b), a common distribution overlap is shared across all clients. Meanwhile, each client owns specific knowledge that cannot be leveraged by others.

Formally, each client associates with a specific data manifold $\mathcal{M}^i$, and the overlapped region of the manifold is shared across all clients, i.e.,

$$\forall\, i \in \{0, 1, ..., N-1\},\ s.t.,\ z \in D^i \subset \mathcal{M}^i\ and\ z \sim p_z^i(z),$$
$$\forall\, i,j,k \in \{0, 1, ..., N-1\}\, (i \neq j \neq k),\ s.t.,\ \mathcal{M}^i \cap \mathcal{M}^j \neq \emptyset\ and\ \mathcal{M}^i \cap \mathcal{M}^j \subset \mathcal{M}^k. \quad (4)$$

Compared with the previous assumptions above, we study a more general scenario *fragmented sharing*, where the shared distribution overlaps are scattered among the clients. Besides, these overlaps are inconsistent across all clients as shown in Figure 2 (c).

**Fragmented sharing.** The local data $z \in D^i$ are sampled from the local manifold $\mathcal{M}^i$ in a particular density $p_z^i(z)$, and there exist overlaps among data manifolds, i.e.,

$$z \in \mathcal{M}^i \subset \mathbb{R}^d, \quad z \sim p_z^i(z) \quad (5a)$$
$$\exists\, i, j \in \{0, 1, ..., N-1\},\ s.t.,\ \mathcal{M}^i \cap \mathcal{M}^j \neq \emptyset. \quad (5b)$$

where $d$ in Eq.(5a) is the dimension of $z$. Eq.(5b) implies the shared overlaps may not be consistent across all clients, e.g., $\cap_{i=0}^{N-1}\mathcal{M}^i = \emptyset$.

## 4 METHODOLOGY

### 4.1 PRELIMINARY: NORMALIZING FLOW

**Normalizing flow.** The generative method NF achieves exact likelihood estimation through an invertible transformation from a known distribution to a complex target distribution. Given a target dataset $D = \{z_0, z_1, ..., z_{n-1}\}, z_i \in \mathbb{R}^d$ and a base variable $e \in \mathbb{R}^d$ with a known density $p_e(e)$, classic NF methods learn a diffeomorphism $f : e_i = g(z_i)$ which maps $p_z$ to the density $p_e$:

$$p_z(z) = p_e\,(g(z))\,|\det J_g\,(g(z))|^{-1}, \quad (6)$$

where $\det J_g\,(g(z)) \in \mathbb{R}^{d \times d}$ denotes the Jacobian matrix evaluated at $g(z)$. Since $g$ is bijective, it is trackable and Eq.(6) could be effectively computed. By fitting the dataset $D$, the approximated distribution $p_z'(z)$ is optimized through a pushforward operation. To enhance the scalability of $g$, one could compose several diffeomorphisms $g = g_{n-1} \circ \cdots \circ g_1 \circ g_0$ for a larger model capacity.

### 4.2 AN OVERVIEW OF PCFL

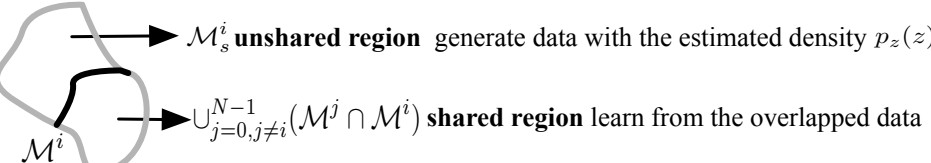

Figure 3: Two meanings of Precision Collaboration. (1) for the shared region of the local manifold $\mathcal{M}^i$, PCFL precisely learns from the overlapped data of other clients; (2) for the unshared region of $\mathcal{M}^i$, PCFL generates synthetic data with the exact density $p_z^i(z)$.

Learning an optimal personalized model $f^i$ for the $i$-th client expects a sufficient utilization of the shared overlaps with other clients. However, due to privacy concerns, one cannot identify these overlaps with direct access to the raw data. We suggest leveraging the overlaps via the learned data manifold to prevent privacy leakage. As shown in Figure 3, in general, our proposed precisely collaborative learning scheme contains:

- for the shared overlaps in other clients, we aim to precisely learn from the shared overlaps identified by the local data manifold $\mathcal{M}^i$;

- for the remaining unshared region of $\mathcal{M}^i$, we expect to advance models with the generated synthetic data from $\mathcal{M}^i$ in an optimal sampling density.

### 4.3 Precision Collaboration I: Learning from the Shared Overlaps

From Figure 1 (a), different clients could share different distribution overlaps, and the distribution overlaps associate with the overlapped region of the local data manifolds. While the data manifold of local clients is mostly agnostic and hardly be inferred by limited local data, we propose to learn the global data manifold with the data on all clients. In this way, all data are utilized and contribute to the manifold inference.

Figure 4: Illustrations of the manifold learning via a NF method. For a complex distribution $p_z(z)$, we learn a tractable injective chart $g \circ h$, which models $p_z(z)$ to a simple distribution $p_{u'}(u')$.

**Learn the global manifold.** The data $z \in D$ are usually supported on an unknown lower-dimensional manifold $\mathcal{M}$. In our realization, we propose to use a NF method to learn the global manifold $\mathcal{M}^g$. Given the data $z$, a bijective transformation $g_\theta$ is used to obtain the latent representation $e \in E$,

$$e = g_\theta(z), \quad where \quad z = g_\theta^{-1}(e), \tag{7}$$

which avoids the risk of information loss during encoding.

While a classical NF requires the latent variable $e \in E$ to have the same dimension with the data space $Z$, following the design in (Brehmer & Cranmer, 2020), we separate the latent space $E = U \times V$ as shown in Figure 4, where $U = \mathbb{R}^{d'}$ denotes the coordinates on the manifold. $V = \mathbb{0}^{d-d'}$ denotes the remaining coordinates, which are the directions orthogonal to the manifold.

To model the density $p_u(u)$, we transform the variable $u$ to the variable $u'$ with the given density $p_{u'}(u')$ using a bijective model $h_\phi$:

$$u' = h_\phi(u), \quad where \quad u = \text{Split}(e), \tag{8}$$

where $\text{Split}(e)$ denotes deleting the $d - d'$ dimensional $\mathbb{0}$ vector from $e$, and $\text{Pad}(u)$ denotes the inverse operation. Please note that in the rest of this paper we will use $g_\theta^*$ to denote $\text{Split} \circ g_\theta$ and $g_\theta^{*-1}$ denotes $g_\theta^{-1} \circ \text{Pad}$.

After the training of the model with the parameters $\theta$ and $\phi$, we learn a diffeomorphism from the data $z \sim p_z(z)$ to a lower dimension space $u' \sim p_{u'}(u')$ with the encoder $h_\phi \circ g_\theta^*$. This means that we transform the original data manifold $\mathcal{M}^g$ to the projected data manifold $U'$. Note that the decoder is the inverse of the encoder. The data is reconstructed given the latent variable $u' \in U'$, i.e., $z = g_\theta^{*-1} \circ h_\phi^{-1}(u')$.

Following the work in (Brehmer & Cranmer, 2020), we train $g_\theta$ and $h_\phi$ by a two-stage optimization framework. In particular, we first train $g_\theta$ to obtain the projection onto the manifold by minimizing the reconstruction error. Then, we optimize $h_\phi$ to approximate the density by maximizing the likelihood (Brehmer & Cranmer, 2020). More implementation details could be found in Appendix due to the page limit.

**Determine the local manifold.** A local data manifold $\mathcal{M}^i$ should contain the local data $D^i$. Considering that the original global manifold $\mathcal{M}^g$ and the latent space $U'$ ($U' = \mathbb{R}^{d'}$) is topologically equivalent, we propose to approximate the local manifold with the projected representation:

$$\mathcal{M}^i = g_\theta^{*-1} \circ h_\phi^{-1}(\overline{U'^i}), \ where \ U'^i = \left\{ h_\phi \circ \ g_\theta^*(x_j^i) \right\}_{j=1}^{n^i}, \tag{9}$$

where $U'^i$ denotes the set of the samples transformed to $U'$ from $D^i$, and $\overline{U'^i}$ is called the projected local data manifold, which is computed as the convex hull of $U'^i$.

Note that $U'^i$ may have a clustered structure. In the realization, we could also firstly cluster the $U'^i$. The union of the convex hull of all clusters is the projected local data manifold, and the original local data manifold ($\mathcal{M}^i$) could be obtained by $\mathcal{M}^i = g_\theta^{*-1} \circ h_\phi^{-1}(\overline{U'^i})$.

**Identify the data overlaps from other clients.** Since we cannot determine the data overlaps directly because of privacy concerns, we propose to identify the overlaps using the learned local manifolds.

Note that the data overlaps correspond to the overlaps of the data manifolds. For example, suppose $D^{i,j}$ is a subset of $D^i$, if $D^{i,j}$ lies on $\mathcal{M}^i$, $D^{i,j}$ is the data overlap between $D^i$ and $D^j$. From Eq.(9), $\mathcal{M}^i$ is reconstructed by $g_\theta^{*-1} \circ h_\phi^{-1}$ with $\overline{U'^i}$. Therefore, $D^{i,j}$ could be identified as follows:

$$D^{i,j} = \left\{ z_k^j | h_\phi \circ g_\theta^*(z_k^j) \in \overline{U'^i}, k = 1, ..., n^j \right\} \subset \mathcal{M}^i. \tag{10}$$

From Eq.(10), the data overlap $D^{i,j}$ is the subset in which each sample is transformed in the projected local data manifold $\overline{U'^i}$.

By learning from the overlaps identified from other clients, we have the following objective,

$$\min_{f_i \in \mathcal{F}} \frac{1}{n^i} \sum_{j=1}^{n^i} \ell(f(x_j^i), y_j^i) + \alpha \cdot \frac{1}{N-1} \sum_{k=0, k \neq i}^{N-1} \mathbb{E}_{(x^k, y^k) \in D^{i,k}}(\ell(f(x^k), y^k)), \tag{11}$$

where $\alpha > 0$ is the regularization parameter, which controls the trade-off between the risk on the $i$-th client and other clients.

## 4.4 PRECISION COLLABORATION II: LEARNING WITH AN OPTIMAL SAMPLING DENSITY

In Sec.4.3, we learn personalized models from the data overlaps between clients. However, the model performance on the unshared data cannot be improved by collaborating with others. The specific region $\mathcal{M}_s^i$ has no overlap with others, which is formulated as

$$\mathcal{M}_s^i = g_\theta^{*-1} \circ h_\phi^{-1}(\overline{U_s'^i}), \ where \ \overline{U_s'^i} = \overline{U'^i} - \cup_{j=0, j \neq i}^{N-1}(\overline{U'^j} \cap \overline{U'^i}), \tag{12}$$

We propose to advance the model by generating data sampled from the local manifold $\mathcal{M}^i$.

While an arbitrary sampling density could generate data $D'^i$ deviated from the local distribution $d(D'^i, D^i) > \epsilon$, this could induce bias to the learned model. An optimal utilization of the synthetic data expects a sampling density close to $p_z^i(z)$. Therefore, we propose to sample from $\mathcal{M}^i$ with the exact estimation of $p_z^i(z)$.

**Exact likelihood estimation.** Note that we learn the manifold by applying a normalizing flow framework, which achieves the exact likelihood estimation simultaneously.

Since we learn the global data manifold, the global data density $p_z^g(z)$ is transformed to $p_{u'}(u')$. For the local data density $p_z^i(z)$, we have the following proposition.

**Proposition 1.** *(proof in Appendix) For any data point $z \in \mathcal{M}_s^i$, the local density $p_z^i(z)$ satisfies*

$$p_z^i(z) = c \cdot p_{u'}(h_\phi \circ g_\theta^*(z)) \left| \det J_{h_\phi}(h_\phi \circ g_\theta^*(z)) \right|^{-1} \left| \det \left[ J_{g_\theta^*}^T(g_\theta^*(x)) J_{g_\theta^*}(g_\theta^*(z)) \right] \right|^{-\frac{1}{2}}, \tag{13}$$

*where $c$ is a proportionality constant, and $J_{h_\phi}$ and $J_{g_\theta^*}$ are the Jacobian matrix of $h_\phi$ and $g_\theta^*$, respectively.*

From the Proposition 1, to sample $z \in \mathcal{M}_s^i$ in the density $p_z^i(z)$, we could firstly sample $u' \sim p_{u'}(u')$ and choose $u' \in \overline{U_s'^i}$ defined in Eq.(12). Then we transform the sampled $u'$ to the data space by $z = g_\theta^{*-1} \circ h_\phi^{-1}(u')$. The final objective is as follows

$$\min_{f_i \in \mathcal{F}} \frac{1}{n^i} \sum_{j=1}^{n^i} \ell(f(x_j^i), y_j^i) + \alpha \cdot \frac{1}{N-1} \sum_{k=0, k \neq i}^{N-1} \mathbb{E}_{(x^k, y^k) \in D^{i,k}}(\ell(f(x^k), y^k)) + \beta \cdot \mathbb{E}_{(x,y) \sim p_z^i(z)} \ell(f(x), y), \tag{14}$$

where the sampled $(x, y)$ in the third term satisfies $(x, y) \in \mathcal{M}_s^i$.

## 4.5 MORE DISCUSSION ON PCFL

From the objective formulated in Eq.(14), PCFL advances a generative framework for efficiently collaborative learning. In addition to the properties summarized as follows, more discussions could be found in Appendix,

- **Scalability:** Since PCFL achieves a smarter utilization of data from other clients, it is pluggable for other FL algorithms. Experiments in Sec. 5 also verify that PCFL could benefit other SOTA baselines;
- **Similarity metric:** PCFL identifies the overlaps among federated networks, which inspires a novel metric for measuring the similarity between clients. More information about it could be found in Appendix.

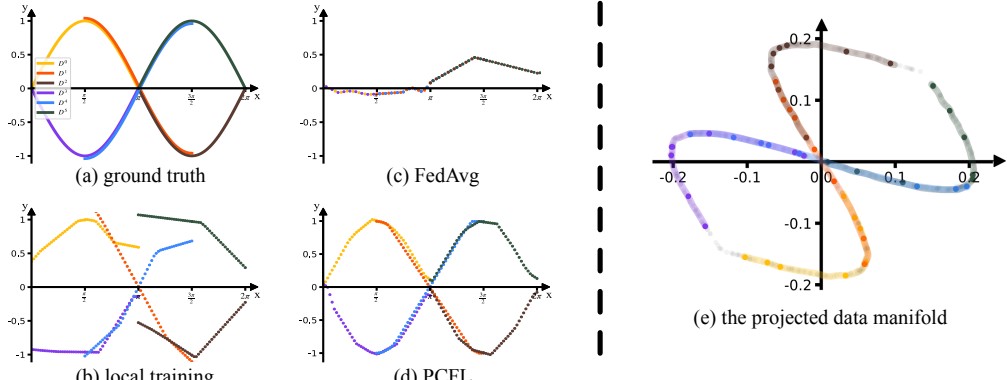

Figure 5: Illustrations of the synthetic experiments. (a) the learning tasks of the six clients; (b), (c) and (d) are the performance of the models learned by local training, FedAvg and PCFL; (e) the learned projected global data manifolds. The points denote the samples from different clients. The colored lines denote the identified local manifolds.

## 5 EXPERIMENTS

We intuitively show the motivation of our method by conducting experiments on synthetic data. We compare our method with various baselines on a wide range of benchmark datasets, including image and tabular datasets. More importantly, the practicability of our method is validated in a real-world clinical federated scenario on eICU dataset (Pollard et al., 2018). The source codes are made publically available at https://github.com/pcfl/pcfl.

### 5.1 SYNTHETIC EXPERIMENTS

**Synthetic data.** Suppose there are 96 clients: $D^i, i \in \{1, 2..., 96\}$. The data points $z = \{x, y\}$ is generated from two objectives $y = sin(x) + \epsilon$ or $y = -sin(x) + \epsilon$ shown in Figure 5 (a), where $\epsilon \sim \mathcal{N}(0, 0.1)$ denotes label noise.

**Fragmented data overlaps.** To generate heterogeneous and overlapped local data, we sample $x$ from the overlapped ranges. In particular, we separate the input space $X$ into four intervals $[0, \frac{\pi}{2}]$, $[\frac{\pi}{2}, \pi]$, $[\pi, \frac{3\pi}{2}]$ and $[\frac{3\pi}{2}, 2\pi]$, and each client randomly chooses two different intervals to sample data. To create conflicting learning tasks, the label of the selected 48 clients is calculated by $y = sin(x) + \epsilon$, and the label of the remaining 48 clients is calculated by $y = -sin(x) + \epsilon$.

| Table 1: CIFAR10 | |
| --- | --- |
| Methods | ACC (%) |
| local | $68.9_{\pm 1.1}$ |
| FedAvg | $72.3_{\pm .5}$ |
| FedProx | $71.5_{\pm .8}$ |
| Fed-MTL | $68.4_{\pm 2.2}$ |
| PerFedAvg | $67.3_{\pm .1}$ |
| LG-FedAvg | $69.2_{\pm .3}$ |
| FedPer | $82.2_{\pm .9}$ |
| FedRep | $82.2_{\pm 1.8}$ |
| APFL | $64.5_{\pm 3.7}$ |
| L2GD | $10.0_{\pm .0}$ |
| Ditto | $82.1_{\pm .2}$ |
| PCFL | $\mathbf{87.3}_{\pm 1.2}$ |

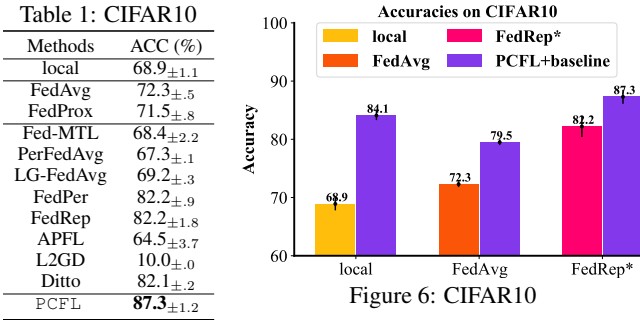

Figure 6: CIFAR10

In this setting, learning a global model for all clients could hurt the model performance as there are two conflicting learning tasks as shown in Figure 5 (c). The best way of collaborative learning for each client is identifying the data overlaps which are sampled from the identical objective with the same intervals. For example, $D^0$ consists of the data sampled from $[0, \frac{\pi}{2}]$ and $[\frac{\pi}{2}, \pi]$, while $D^1$ consists of the data sampled from $[\frac{\pi}{2}, \pi]$ and $[\pi, \frac{3\pi}{2}]$ shown in Figure 5 (a). Learning an optimal model for $D^0$ needs to precisely identify the data overlap sampled from $[\frac{\pi}{2}, \pi]$ in $D^1$. From Figure 5 (e), PCFL efficiently obtains local data manifolds and identifies the data overlaps between clients. Therefore, PCFL learns a better model by precision collaboration which maximizes the benefits and avoids potential negative transfer from other clients as shown in Figure 5 (d).

**Datasets.** We adopt three benchmark image datasets: CIFAR10 (Krizhevsky et al., 2009), FEM-NIST (Caldas et al., 2018), CelebA (Liu et al., 2015), and a tabular dataset *Adult* (Kohavi et al., 1996). We create the federated environment with data heterogeneity for CIFAR10 by randomly

allocating several classes to each client following the work (McMahan et al., 2017). We use $K$ to denote the number of clients and $S$ to denote the number of classes in each client. For CIAFR10, $K = 150, S = 5$ means there are 150 clients and each client contains 5 classes of images. For FEMNIST which has 10 classes of handwritten letters, we consider the setting of $K = 200, S = 5$. The number of samples in each client is determined according to a log-normal distribution (Li et al., 2019a). The task on CelebA is to classify whether the celebrity in the image is smiling (Li et al., 2021b). There are 545 clients and 21 samples per client in average. For the tabular dataset *Adult*, the task is to predict whether an individual's income is beyond 50K/year based on some census features, including age, race, workclass, etc. Following the setting in (Mohri et al., 2019), all individuals are split into two clients. One is PhD client and the other is non-PhD client.

## 5.2 BENCHMARK EXPERIMENTS

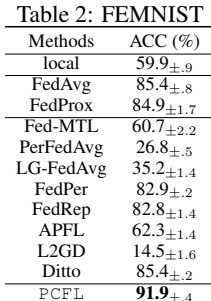

Table 2: FEMNIST

| Methods | ACC (%) |
|---|---|
| local | $59.9_{\pm.9}$ |
| FedAvg | $85.4_{\pm.8}$ |
| FedProx | $84.9_{\pm1.7}$ |
| Fed-MTL | $60.7_{\pm2.2}$ |
| PerFedAvg | $26.8_{\pm.5}$ |
| LG-FedAvg | $35.2_{\pm1.4}$ |
| FedPer | $82.9_{\pm.2}$ |
| FedRep | $82.8_{\pm1.4}$ |
| APFL | $62.3_{\pm1.4}$ |
| L2GD | $14.5_{\pm1.6}$ |
| Ditto | $85.4_{\pm.2}$ |
| PCFL | $\mathbf{91.9_{\pm.4}}$ |

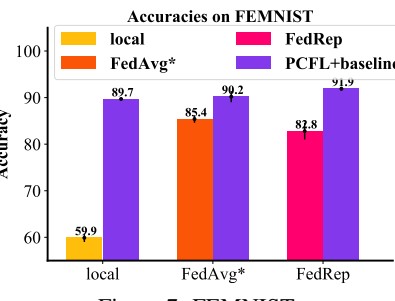

Figure 7: FEMNIST

**Baselines.** We compare our method with various baseline, including global and personalized methods. Global baselines include: 1) Fe-dAvg (McMahan et al., 2017); 2) FedProx (Li et al., 2020). Person-alized baselines include: 1) Fed-MTL (Smith et al., 2017); 2) PerFe-dAvg (Fallah et al., 2020); 3) LG-FedAvg (Liang et al., 2020); 4) Fed-Per (Arivazhagan et al., 2019); 5) Fe-dRep (Collins et al., 2021); 6) APFL (Deng et al., 2020); 7) L2GD (Hanzely & Richtárik, 2020); 8) Ditto (Li et al., 2021b).

**Experimental Results.** The accuracy of all methods on CIFAR10 dataset are shown in Table 1[1]. PCFL outperforms all baselines on this classification task. Since each client has insufficient data samples ($n^i = 333$), FedAvg (72.3%) learning from all data has a better performance compared with local (68.9). From Table 1, FedRep (82.2%) surpasses other baselines by learning a global feature extractor. As a pluggable method, PCFL could be used to enhance the performance of other art methods. From Figure 6, PCFL improves the performance of FedRep by 5.1%, which indicates that PCFL effectively identifies the informative knowledge from others.

Similar phenomena could also be found in the experimental results on FEMNIST. FedAvg achieves a better performance compared with other baselines because of the relatively slighter heterogeneity. PCFL also outperforms all baselines on both learning tasks. Moreover, PCFL successfully boosts the models learning on the three baselines as shown in Figure 7. More experimental results could be found in Appendix.

Table 3: Adult

| Methods | average | non-PhD | PhD |
|---|---|---|---|
| local | $83.3_{\pm.1}$ | $83.4_{\pm.1}$ | $70.2_{\pm.4}$ |
| FedAvg | $83.3_{\pm.3}$ | $83.4_{\pm.3}$ | $72.9_{\pm.2}$ |
| FedProx | $83.3_{\pm.1}$ | $83.5_{\pm.1}$ | $71.8_{\pm.5}$ |
| Fed-MTL | $83.3_{\pm.2}$ | $83.4_{\pm.2}$ | $69.1_{\pm.4}$ |
| PerFedAvg | $83.4_{\pm.2}$ | $\mathbf{83.6_{\pm.2}}$ | $70.2_{\pm2.0}$ |
| APFL | $83.0_{\pm.1}$ | $83.2_{\pm.1}$ | $69.1_{\pm.3}$ |
| L2GD | $70.4_{\pm.0}$ | $70.4_{\pm.0}$ | $71.8_{\pm0.6}$ |
| Ditto | $83.4_{\pm.2}$ | $83.5_{\pm.2}$ | $75.7_{\pm.9}$ |
| PCFL | $\mathbf{83.5_{\pm.1}}$ | $\mathbf{83.6_{\pm.1}}$ | $\mathbf{77.3_{\pm.7}}$ |

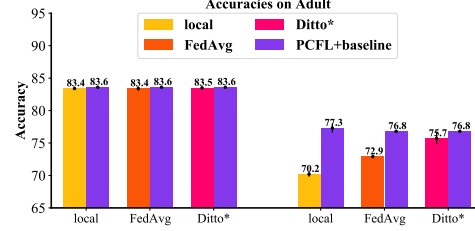

Figure 8: Adult

We also evaluate the availability of our proposed method on a tabular data Adult, and the results on the two clients are shown in Table 3. Since the classifier on Adult is one-layer MLP, LG-FedAvg, FedPer and FedRep degrade into local training. Compared with PhD client which has 413 training samples, non-PhD has more than 30000 training samples. From Table 3, all methods achieve similar performance on non-PhD client. Because of the severe distribution discrepancy, naive averaging may not lead to optimal accuracy on non-PhD client. From Figure 8, by leveraging the favorable data

---

[1]We show the best performance of PCFL in all tables. For all figures in experiments, we show the results of Local, FedAvg and the best baseline.

in non-PhD client and the learned manifold of PhD client, `PCFL` substantially improves the model performance of local training ($\uparrow 7.1\%$).

## 5.3 REAL DATA EXPERIMENTS

To further verify the practicability of `PCFL`, we conduct experiments on a real-world clinical dataset eICU (Pollard et al., 2018). eICU contains the patients to ICUs with hospital information. Naturally, hospitals located in different areas are local clients as in (Cui et al., 2022), where the patient data are kept confidential. We preprocess the data following the work (Sheikhalishahi et al., 2019) and each data spans a 1-hour window. The task is to predict in-hospital mortality of each instance using the 48-hour monitoring data.

Table 4: eICU

| Methods | AUC(%) |
|---------|--------|
| local | $73.7_{\pm1.4}$ |
| FedAvg | $73.2_{\pm.5}$ |
| FedProx | $78.2_{\pm.2}$ |
| Fed-MTL | $77.2_{\pm1.6}$ |
| PerFedAvg | $73.8_{\pm.3}$ |
| LG-FedAvg | $74.5_{\pm.2}$ |
| FedPer | $74.3_{\pm.7}$ |
| FedRep | $74.1_{\pm1.2}$ |
| APFL | $68.3_{\pm.8}$ |
| L2GD | $72.0_{\pm.6}$ |
| Ditto | $78.3_{\pm.1}$ |
| `PCFL` | $\mathbf{80.0}_{\pm.6}$ |

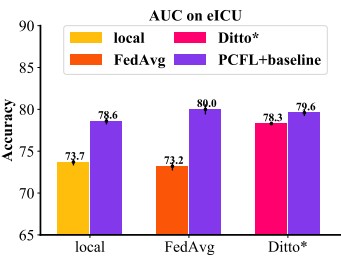

Figure 9: eICU

In the experiments, we randomly select 14 hospitals in the federated network. We use a Bi-LSTM to implement this binary classification. We use AUC as the metric due to the severe label imbalance (more than $90\%$ samples have negative labels).

From Table 4, `PCFL` still maintains the best model utility. FedAvg achieves a comparable performance compared with local. While different hospitals own different populations which could result in data heterogeneity, Ditto learns a robust personalized model and achieves better performance ($78.3\%$). The results shown in Figure 9 prove that `PCFL` could also benefit the baselines in real-world scenarios.

## 5.4 ABLATION STUDIES

While `PCFL` formulated in Eq.(14) consists three terms: 1). loss of local training; 2). loss on the identified overlapped data from other clients; 3). loss on sampled data from the manifold, to analysis the effect of each component, we conduct ablation studies on several datasets. More ablation studies and implementation details could be found in Appendix.

When $\alpha = 0$, only the local data and sampled data from local manifolds are used for model learning. When $\beta = 0$, only the local data and the identified overlapped data from other clients are utilized for model learning. Experiments in Table 5 demonstrate that 1). both the identified distributional overlaps ($\beta = 0$) and the data sampled with a learned distribution density ($\alpha = 0$) facilitate the model learning; 2). the identified overlaps ($\beta = 0$) could achieve more performance gain than the generated data ($\alpha = 0$).

Table 5: Ablation studies of `PCFL` formulated Eq.(14).

| Dataset | local | $\alpha = 0$ | $\beta = 0$ | `PCFL` |
|---------|-------|--------------|-------------|--------|
| CIFAR10 | $68.9_{\pm1.1}$ | $76.7_{\pm.3}$ | $79.4_{\pm.7}$ | $\mathbf{84.1}_{\pm.8}$ |
| FEMNIST | $59.9_{\pm.9}$ | $75.5_{\pm1.1}$ | $82.7_{\pm.8}$ | $\mathbf{89.7}_{\pm.2}$ |
| CelebA | $69.3_{\pm1.1}$ | $82.6_{\pm2.7}$ | $80.8_{\pm.9}$ | $\mathbf{85.8}_{\pm1.1}$ |
| eICU | $73.7_{\pm.4}$ | $76.6_{\pm.3}$ | $77.4_{\pm.5}$ | $\mathbf{78.6}_{\pm.4}$ |

## 6 CONCLUSION

In this paper, we propose a precise collaboration framework `PCFL` for a more general FL learning scenario, where the fragmented and shared knowledge is distributed among other clients. Experiments on benchmark datasets and a real-world clinical dataset verify the superiority of our method because of optimal and precise utilization of the shared information. Our framework determines the overlaps between clients, which suggests several attractive topics, such as identifying malicious clients or noisy data in the federated network. Moreover, `PCFL` encourages a novel similarity metric stated in Sec. C.1. This metric could be used to provide incentives or impose charges on each client, to promote the practicality of FL in real-world applications.

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

## A  THEORETICAL PROOFS

### A.1  PROOF OF PROPOSITION 1

Suppose there is a smooth and injective mapping $g_\theta^* : \mathbb{R}^d \to \mathbb{R}^{d'}$ with $d' \leq d$, $U \in \mathbb{R}^{d'}$ is the latent variable and has $Z := g_\theta^{*-1}(U)$. From differential geometry (Krantz & Parks, 2008), we have

$$p_z(z) = p_u\left(g_\theta^*(z)\right) \left| \det J_{g_\theta^*}^\top \left(g_\theta^*(z)\right) J_{g_\theta^*}\left(g_\theta^*(x)\right) \right|^{-1/2}. \tag{15}$$

Suppose there is a smooth and bijective mapping $h_\phi : \mathbb{R}^{d'} \to \mathbb{R}d'$, $U' \in \mathbb{R}^{d'}$ is the latent variable which has $U := h_\phi(U')$. We have

$$p_u(u) = p_{u'}\left(h_\phi(u)\right) \left| \det J_{h_\phi}(h_\phi(u)) \right|^{-1}. \tag{16}$$

According to the chain rule, combining Eq.(15) and Eq.(16), we have

$$p_z(z) = p_{u'}\left(h_\phi \circ g_\theta^*(z)\right) \left| \det J_{h_\phi}(h_\phi \circ g_\theta^*(z)) \right|^{-1} \left| \det J_{g_\theta^*}^\top \left(g_\theta^*(z)\right) J_{g_\theta^*}\left(g_\theta^*(z)\right) \right|^{-1/2}. \tag{17}$$

Since we learn a global manifold $\mathcal{M}^g$ with the data from all clients, the density of the data from all clients $p_z^g(z)$ is approximated in Eq.(17). From the definition of $\mathcal{M}_s^i$ in Eq.(12), if there is a data point $z \in \mathcal{M}_s^i$, $z$ will cannot be sampled from any other manifolds $\mathcal{M}^j$ ( $j \neq i$ ) but $\mathcal{M}^i$, i.e.,

$$\forall z \in \mathcal{M}^g, \quad s.t., \quad z \notin \mathcal{M}^j (j \neq i) \, if \, z \in \mathcal{M}_s^i. \tag{18}$$

Therefore, we have

$$p_z^g(z, z \in \mathcal{M}_s^i | z \in \mathcal{M}^i) = \frac{p_z^g(z, z \in \mathcal{M}_s^i, z \in \mathcal{M}^i)}{p_z^g(z \in \mathcal{M}^i)} = \frac{p_z^g(z, z \in \mathcal{M}_s^i)}{p_z^g(z \in \mathcal{M}^i)} = \frac{1}{p_z^g(z \in \mathcal{M}^i)} \cdot p_z^i(z, z \in \mathcal{M}_s^i). \tag{19}$$

Combining with Eq.(17), for $z \in \mathcal{M}_s^i$, we have

$$p_z^i(z) = c \cdot p_{u'}\left(h_\phi \circ g_\theta^*(z)\right) \left| \det J_{h_\phi}(h_\phi \circ g_\theta^*(z)) \right|^{-1} \left| \det J_{g_\theta^*}^\top \left(g_\theta^*(z)\right) J_{g_\theta^*}\left(g_\theta^*(z)\right) \right|^{-1/2}, \tag{20}$$

and Proposition 1 holds.

## B  PIPELINE OF OUR FRAMEWORK PCFL

The pipeline of the global manifold learning $\mathcal{M}^g$ is elaborated in Algorithm 1. We learn a global manifold model in the federated learning setting. There are two phases of training. Firstly, only the parameters of $g_\theta$ are updated as in Line 5-7. Then the parameters of $h_\phi$ are updated as in Line 9-10. The learned manifold model $g_\theta^* \circ h_\phi$ is utilized in our framework PCFL, whose pipeline is elaborated in Algorithm 2. To begin with, the local manifolds of clients are extracted based on Eq.(9) and the distribution overlaps are calculated based on Eq.(10). Since only the borders of convex hulls are exchanged, there is no leakage of sensitive information. The data from the overlapped distribution of other clients are used to train the models. They are utilized by transmitting the average gradients through the server as in Line 7-14.

---

**Algorithm 1** Learn the global manifold in the federated learning framework

---

**Input:** epoch $T_m$, batch size $B_m$, initial manifold model $\mathcal{M}^g$ with the parameters $\theta$ and $\phi$.

1: **for** $t = 0, ..., T_m - 1$ **do**
2:   randomly select a subset of clients $S_t$
3:   **for** client $D^i \in S_t$ in parallel **do**
4:     draw mini-batch $\mathbf{z}^i : z_{t_1}^i, ..., z_{t_{B_m}}^i \sim D^i$
5:     **if** $t < T_m/2$ **then**
6:       calculate the loss: $\frac{1}{B_m} \sum_{i=1}^{B_m} \| z^i - g_\theta^{-1}(g_\theta(z^i)) \|$;
7:       then calculate the gradients of loss with respect to parameters $\theta$;
8:     **else**
9:       calculate the loss: $-\frac{1}{B_m} \sum_{i=1}^{B_m} \left( \log p_{u'}(h_\phi \circ g_\theta^*(z^i)) - \log \det J_h(h_\phi \circ g_\theta^*(z^i)) \right)$;
10:      then calculate the gradients of loss with respect to parameters $\phi$;
11:     **end if**
12:   **end for**
13:   **Server** aggregates the gradients of selected clients and update the parameters $\theta$ and $\phi$.
14: **end for**
15: **Output:** the learned manifold model $g_\theta$ and $h_\phi$.

---

**Algorithm 2** Federated learning framework `PCFL`

---

**Input:** epoch $T$, batch size $B$, initial models $\{f^0, ..., f^{N-1}\}$, hyperparameters $\alpha$ and $\beta$;

1: all the clients determine the local manifold $\mathcal{M}^i$ and $\overline{U'^i}$ based on Eq.(9), and send $\overline{U'^i}$ to the **Server**.
2: the **Server** calculates the overlaps of $\overline{U'^i}$ between clients, calculates $\overline{U_s'^i}$ based on Eq.(12), and sends them to each client;
3: **for** $t = 0, ..., T - 1$ **do**
4:   randomly select a subset of clients $S_t$,
5:   the selected clients send their local models to the **Server**;
6:   **for** client $D^i \in S_t$ in parallel **do**
7:     draw mini-batch $(x^i, y^i) \sim D^i$;
8:     calculate the loss $\mathbb{E}_{(x^i, y^i) \in D^i}(\ell(f_i(x^k), y^k)) + \beta \cdot \mathbb{E}_{(x,y) \in p_z^i(z)} \ell(f_i(x), y)$, and update the model $f^i$ using the gradients of loss;
9:     **for** $k = 0, ..., N, k \neq i$ **do**
10:       draw mini-batch $(x^k, y^k) \sim D^{i,k}$
11:       calculate the loss $\alpha \cdot \mathbb{E}_{(x^k, y^k) \in D^{i,k}} \ell(f_i(x^k), y^k)$, and update the model $f^i$ using the gradients of loss;
12:     **end for**
13:     the **Server** aggregates the parameters of $f^i$ from other clients and send the average to the $i$-th client;
14:     then the $i$-th client $D^i$ updates the model $f^i$ with the received parameters and local gradients.
15:   **end for**
16: **end for**
17: **Output:** the learned personalized models $\{f^0, ..., f^{N-1}\}$.

---

## C    MORE DISCUSSIONS ABOUT PCFL

### C.1    A NEW METRIC OF CLIENT SIMILARITY

Our framework `PCFL` inspires a novel metric for measuring the similarity between local clients. For example, suppose the $i$-th client and $j$-th client has the identical local manifold $\mathcal{M}^i = \mathcal{M}^j$, the similarity between clients is close to 1. On the contrary, if the two local manifolds are disjoint $\mathcal{M}^i \cap \mathcal{M}^j = \emptyset$, the measured similarity should be 0. In particular, we propose to measure the similarity as the Intersection of Union (IoU) of the projected local manifold,

$$S(D^i, D^j) = \text{IoU}(\overline{U'^i}, \overline{U'^j}). \tag{21}$$

**A communication-efficient client-level collaboration.** Our proposed metric allows efficient collaborator identification which reduces the communication and computation overhead. For example, we require $D^i$ to collaborate with certain clients who have a higher client similarity:

$$\min_{f \in \mathcal{F}} \frac{1}{\sum_{k=0, S(D^i, D^k) \geq \epsilon}^{N-1} n^k} \sum_{k=0, S(D^i, D^k) \geq \epsilon}^{N-1} \sum_{i=1}^{n^k} l\left(f\left(x_i^k\right), y_i^k\right), \tag{22}$$

where $\epsilon \geq 0$ is a pre-defined threshold. Note that the objective in Eq.(22) is different from clustered FL methods. Clustered FL methods learn a common model for each cluster while Eq.(22) learns a personalized model for each client. Experimental results shown in Sec. E.2 verify that this method achieves a comparable performance while reducing communication and computation overhead.

Previous work has explored the problem of identifying similar datasets in a graph network for downstream learning tasks (Hallac et al., 2015). In particular, Jung (2020) formulate the learning from distributed local datasets as a convex optimization problem, and proposes to cluster the local datasets according to the learned parameters. Jung & Tran (2019) extend network lasso methods in regression tasks under a clustering assumption. These cluster-based methods could be applied in federated learning with a proper design for privacy-preserving. In our experiments, we use network lasso to cluster the local datasets under the federated setting. In Table 6, we show the comparison of PCFL and the clustered methods. Our method outperforms all cluster-based methods, which demonstrates that a precision identification of overlaps in other clients facilitates model learning. Moreover, an interesting direction is the application of our proposed similarity metric in the graph network. For example, the manifold learning of local datasets in the graph network may also be used for similarity measurement.

Table 6: More experimental results on eICU

| Methods | AUC (%) |
|---|---|
| local | $73.7_{\pm 1.4}$ |
| FedAvg | $73.2_{\pm .5}$ |
| FedProx | $78.2_{\pm .2}$ |
| Fed-MTL | $77.2_{\pm 1.6}$ |
| PerFedAvg | $73.8_{\pm .3}$ |
| LG-FedAvg | $74.5_{\pm .2}$ |
| FedPer | $74.3_{\pm .7}$ |
| FedRep | $74.1_{\pm 1.2}$ |
| APFL | $68.3_{\pm .8}$ |
| L2GD | $72.0_{\pm .6}$ |
| Ditto | $78.3_{\pm .1}$ |
| Clustered FL | $74.7_{\pm .3}$ |
| Network Lasso | $76.3_{\pm .8}$ |
| ours | $80.0_{\pm .6}$ |

## C.2 COMPUTATION COMPLEXITY AND OPTIMIZATION EFFICIENCY

From Algorithm 1 and Algorithm 2, PCFL is realized by a two-staged optimization framework. For the training of the normalizing flow in the first stage, PCFL learns a global model for all clients, which has the computation complexity as FedAvg. For the identification of the manifold overlaps, it has $O(1)$ time complexity as the server only computes it once. For the training of local models in the second stage, PCFL learns a personalized model for each client, which has the computation complexity as other personalized methods. From the above all, PCFL achieves a similar computation complexity as baselines.

A classical NF method requires a fixed dimensionality of the latent space, which is the same as the dimension of the data. In this case, learning such a NF model could bring a huge computation overhead when the data is high-dimensional. In the first phase of our framework, we learn a low-dimensional manifold in a NF method, which significantly reduces the computation overhead.

Our method learns from local data, data overlaps of other clients, and sampled data in the manifold. By precision collaboration, we avoid learning from all data. We make comparisons of run-time consumption with the baselines. The experiments are conducted on the same device NVIDIA GeForce

RTX 2080 Ti. The results on eICU dataset are displayed in Table 7. As a pluggable method, the time consumption of `PCFL` is comparable to the corresponding baselines. Fed-MTL involves computing the correlation of the parameters among all client models, which could result in more computation overhead.

Table 7: Run-time consumption comparisons

| Methods | Run-time consumption |
|---------|---------------------|
| local | 33 min 47 s |
| FedAvg | 57 min 41 s |
| FedProx | 56 min 35 s |
| Fed-MTL | 101 min 12 s |
| PerFedAvg | 79 min 47 s |
| LG-FedAvg | 55 min 27 s |
| FedPer | 57 min 43 s |
| FedRep | 40 min 37 s |
| APFL | 92 min 13 s |
| L2GD | 63 min 4 s |
| Ditto | 71 min 34 s |
| PCFL | 74 min 27 s |

## C.3 PRIVACY PRESERVING

`PCFL` maintains data confidentiality as baselines because there is no shared data between local clients. `PCFL` achieves privacy-preserving as baselines because our framework learns models by communicating model parameters only. Federated learning may need further exploration to maintain data privacy. Some researchers claim there is information leakage when sharing models or gradients (Zhu et al., 2019). To alleviate this issue, there are research proposing to apply other techniques to FL methods, such as differential privacy (Wei et al., 2020), secure multi-party computation, etc. `PCFL` is also compatible with these techniques.

## D ABLATION STUDIES

### D.1 MORE ABLATION STUDIES ABOUT `PCFL`

`PCFL` is pluggable for other FL algorithms. We test local, FedAvg, FedRep and Ditto which are implemented with/without our method as in Table 8 and Table 9. In the datset Adult, all individuals are split into two clients, one of which is PhD client and the other is non-PhD client. The non-PhD client contains 32148 training samples while the PhD client contains 413 samples. Therefore the non-PhD client of Adult can not benefit much from federated learning methods. In other datasets, our method boosts the baselines by large margins.

Table 8: Experiment results of `PCFL` implemented on CIFAR10, FEMNIST, and CelebA (%)

| Dataset | local | PCFL (local) | FedAvg | PCFL (FedAvg) | FedRep | PCFL (FedRep) |
|---------|-------|--------------|--------|---------------|--------|---------------|
| CIFAR10 | $68.9_{\pm 1.1}$ | $84.1_{\pm .8}$ (↑ 15.2) | $72.3_{\pm .5}$ | $79.5_{\pm .5}$ (↑ 7.2) | $82.2_{\pm 1.8}$ | $87.3_{\pm 1.2}$ (↑ 5.1) |
| FEMNIST | $59.9_{\pm .9}$ | $89.7_{\pm .2}$ (↑ 29.8) | $85.4_{\pm .8}$ | $90.2_{\pm 1.2}$ (↑ 4.8) | $82.8_{\pm 1.4}$ | $91.9_{\pm .4}$ (↑ 9.1) |
| CelebA | $69.3_{\pm 1.1}$ | $85.8_{\pm 1.1}$ (↑ 16.5) | $85.2_{\pm 2.1}$ | $89.5_{\pm 2.0}$ (↑ 4.3) | $68.1_{\pm .6}$ | $71.2_{\pm .6}$ (↑ 3.1) |

Table 9: Experiment results of `PCFL` on eICU and Adult (%)

| Dataset | local | PCFL (local) | FedAvg | PCFL (FedAvg) | Ditto | PCFL (Ditto) |
|---------|-------|--------------|--------|---------------|-------|--------------|
| Adult non-PhD | $83.4_{\pm .1}$ | $83.6_{\pm .1}$ (↑ .2) | $83.4_{\pm .3}$ | $83.6_{\pm .1}$ (↑ .2) | $83.5_{\pm .2}$ | $83.6_{\pm .0}$ (↑ .1) |
| Adult PhD | $70.2_{\pm .4}$ | $77.3_{\pm .7}$ (↑ 7.1) | $72.9_{\pm .2}$ | $76.8_{\pm .2}$ (↑ 3.9) | $75.7_{\pm .9}$ | $76.8_{\pm .1}$ (↑ 1.1) |
| eICU | $73.7_{\pm 1.4}$ | $78.6_{\pm .4}$ (↑ 4.9) | $73.2_{\pm .5}$ | $80.0_{\pm .6}$ (↑ 6.8) | $78.3_{\pm .1}$ | $79.6_{\pm .4}$ (↑ 1.3) |

## E EXPERIMENTS AND IMPLEMENTATION DETAILS

### E.1 MORE EXPERIMENTS

The experimental results on CelebA are shown in Table 11 and Figure 10. Moreover, we conduct experiments on FEMNIST on more heterogeneous settings with more clients. We partition the dataset

into 400 clients with the Dirichlet distribution $Dir_{400}(0.1)$ and $Dir_{400}(0.5)$ following the work in (Wang et al., 2019). We compare our method with the baselines, and the results are shown in Table 10. With more clients, each client has fewer training samples. Local method shows poor performance (63.6% in $Dir_{400}(0.5)$). Global methods (FedAvg and FedProx) achieve better performance under a less heterogeneous setting ($Dir_{400}(0.5)$), while the performance of personalized methods degrades. Under two settings ($Dir_{400}(0.1)$ and $Dir_{400}(0.5)$), PCFL outperforms all baselines by identifying the informative overlaps for each client.

Table 10: More experimental results on FEMNIST (Acc %)

| methods | $Dir_{400}(0.1)$ | $Dir_{400}(0.5)$ |
|---|---|---|
| local | $71.8_{\pm.8}$ | $63.6_{\pm.4}$ |
| FedAvg | $69.1_{\pm.5}$ | $80.6_{\pm1.7}$ |
| FedProx | $67.8_{\pm.3}$ | $79.9_{\pm.2}$ |
| Fed-MTL | $81.8_{\pm.9}$ | $60.1_{\pm.4}$ |
| PerFedAvg | $82.4_{\pm1.1}$ | $46.7_{\pm.8}$ |
| LG-FedAvg | $86.8_{\pm.8}$ | $49.5_{\pm.5}$ |
| FedPer | $91.1_{\pm.3}$ | $76.8_{\pm.2}$ |
| FedRep | $91.8_{\pm1.4}$ | $74.6_{\pm.3}$ |
| APFL | $79.9_{\pm.9}$ | $60.9_{\pm.5}$ |
| L2GD | $77.6_{\pm.4}$ | $39.9_{\pm1.5}$ |
| Ditto | $91.7_{\pm.7}$ | $82.9_{\pm.4}$ |
| PCFL | $96.1_{\pm.4}$ | $88.3_{\pm.6}$ |

## E.2 EVALUATION ABOUT THE PROPOSED NOVEL METRIC

Table 11: CelebA

| Methods | ACC (%) |
|---|---|
| local | $69.3_{\pm1.1}$ |
| FedAvg | $85.2_{\pm2.1}$ |
| FedProx | $81.2_{\pm1.2}$ |
| Fed-MTL | $68.2_{\pm.4}$ |
| PerFedAvg | $68.6_{\pm.8}$ |
| LG-FedAvg | $68.4_{\pm1.2}$ |
| FedPer | $68.2_{\pm.5}$ |
| FedRep | $68.1_{\pm.6}$ |
| APFL | $71.4_{\pm.6}$ |
| L2GD | $67.9_{\pm2.0}$ |
| Ditto | $84.5_{\pm.6}$ |
| PCFL | $\mathbf{89.5}_{\pm2.0}$ |

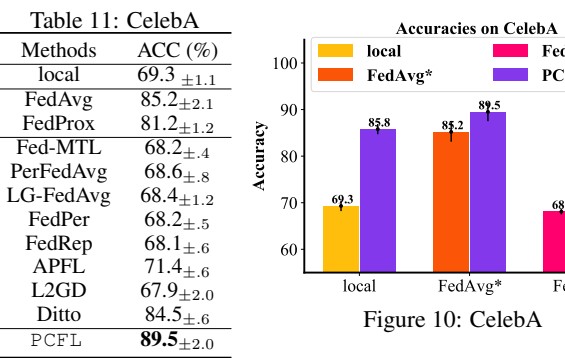

Figure 10: CelebA

In Sec. C.1, we propose a novel metric for measuring the distance between local clients, which could be used for a communication-efficient client-level collaboration. We conduct experiments on eICU dataset, in which we select the most similar 7 clients for each client to learn a personalized model. The experimental results are shown in Table 12.

From Table 12, our method for identifying the collaborators achieves a comparable performance compared with baselines and reduces computation and communication overhead by collaborating with a subset of local clients.

To explore the effect of $\epsilon$ on the performance of the learned models, we set $\epsilon$ by controlling the number of clients to collaborate for each client. There are 14 clients in eICU dataset. We test the number of the collaborator ($C$) to be 1, 3, 5, ... etc. The results are shown in Table ??. When $C = 7$, the learned model achieves the highest AUC (78.0). When $C > 7$, the performance tends to remain unchanged.

## E.3 IMPLEMENTATION DETAILS

Our method is implemented with Pytorch and all experiments are run 5 times to calculate the average results with stds. We use a four-layer MLP for the synthetic experiment, three-layer MLP for FEMNIST, two-layer CNNs for CIFAR10 and CelebA, and a one-layer MLP for Adult. Following the work (Collins et al., 2021), for all the methods we sample 10% of the clients in every global epoch. We train the models for 200 global epochs on FEMNIST, CIFAR10 and CelebA, 50 on Adult. And we train 15 local epochs for FEMNIST, CIFAR10 and Adult in every global epoch, 25 for CelebA. All models are trained with stochastic gradient descent. We use grid search to find the optimal hyperparameters $\alpha$ and $\beta$ in the validation set of each dataset. We set $\alpha = 0.5$, $\beta = 0.5$ for CIFAR10, CelebA, Adult; and set $\alpha = 1$, $\beta = 0.5$ for FEMNIST and eICU. Besides, we test different

Table 12: Experimental results on eICU

| Methods | AUC (%) |
|---|---|
| local | $73.7_{\pm 1.4}$ |
| FedAvg | $73.2_{\pm .5}$ |
| FedProx | $78.2_{\pm .2}$ |
| Fed-MTL | $77.2_{\pm 1.6}$ |
| PerFedAvg | $73.8_{\pm .3}$ |
| LG-FedAvg | $74.5_{\pm .2}$ |
| FedPer | $74.3_{\pm .7}$ |
| FedRep | $74.1_{\pm 1.2}$ |
| APFL | $68.3_{\pm .8}$ |
| L2GD | $72.0_{\pm .6}$ |
| Ditto | $78.3_{\pm .1}$ |
| ours | $78.0_{\pm .1}$ |

Table 13: Experimental results on eICU with adaptive $\epsilon$

| $C$ | AUC (%) |
|---|---|
| 1 | $69.0_{\pm .4}$ |
| 3 | $75.4_{\pm .2}$ |
| 5 | $75.5_{\pm .7}$ |
| 7 | $78.0_{\pm .1}$ |
| 9 | $77.1_{\pm .9}$ |
| 11 | $76.8_{\pm .2}$ |
| 13 | $77.0_{\pm .3}$ |

manifold dimensions $d'$ for each benchmark dataset. We keep $d'$ as small as possible while ensuring reconstruction quality on the validation set. We $d' = 256$ for CIFAR10 and CelebA, $d' = 12$ for FEMNIST, $d' = 32$ for Adult and eICU. For synthetic experiment, the data dimension $d = 3$ and manifold dimension $d' = 2$ since one element of data $z$ identically equals to 0. The source codes are made publically available at https://github.com/pcfl/pcfl.

### E.4 DATASETS

In our experiments, CIFAR10, FEMNIST, CelebA and Adult are all public dataset. For the synthetic experiment, the data point $z = \{x, 0, y\}$ has three elements. We add a zero element to data so that the manifold dimension is smaller than the data dimension, which simulates the situation in real-world datasets. We create the federated environment with data heterogeneity for CIFAR10 and FEMNSIT by randomly allocating several classes to each client following the work (McMahan et al., 2017). For the dataset eICU, we follow the procedure on the website https://eicu-crd.mit.edu and got the approval for the dataset. We follow the data preprocessing as in Sheikhalishahi et al. (2019) and randomly select 14 hospitals as introduced in the main text.

### E.5 COMPUTING RESOURCES

Part of the experiments is conducted on a local server with Ubuntu 16.04 system. It has two physical CPU chips which are Intel(R) Xeon(R) CPU E5-2667 v4 @ 3.20GHz with 32 logical kernels. The other experiments are conducted on a remote server. It has 8 GPUs which are GeForce RTX 2080 Ti.

