# OpenReview forum: "Precision Collaboration for Federated Learning"
_ICLR.cc/2023/Conference — Submitted to ICLR 2023_

### Official Review · Reviewer_XA4j · 2022-10-22

**Confidence:** 4
**Correctness:** 2
**Technical Novelty And Significance:** 2
**Empirical Novelty And Significance:** 3
**Recommendation:** 5

**Clarity, Quality, Novelty And Reproducibility:**

The paper is presented relatively clearly, but the motivation of some parts of the method is unclear. The problem studied in the paper is novel and practical. The method section is missing some descriptions on how to train the model parameters, which harms the reproducibility.

**Strength And Weaknesses:**

Strengths:
1)	This paper studies a more practical and general scenario where the data distribution overlaps across different clients could be fragmented, that is, the informative and ambiguous data shards exist simultaneously in other clients.
2)	The authors proposed to utilize data manifolds to identify the meaningful overlaps and exclude ambiguous information from the clients.
Weaknesses:
1)	In section 4.3, the authors used the normalizing flow to infer the local data manifolds. However, there is no motivation to separate the latent space into two parts. What is the point of doing so?
2)	The authors utilized two bijective transformation including g_{\theta} and h_{\Phi} to transform the original data manifold to the lower projected data manifold. However, the authors did not explain how to learn the parameters of the two transformation in the method and experiment section.
3)	In Equation (16), the authors required client i to collaborate with certain clients who have a higher client similarity. But it is confusing to minimize the loss of the model on other clients’ datasets.
4)	In Section 4.4, the final objective in Equation (14) uses both the data of client i and part of the data of other clients to train the local model. However, this private data should be kept locally on each client. So how did the authors train models with the data at the same time?
5)	The authors should experiment with the sensitivity of the parameters of the proposed method, such as the threshold \epsilon, which has a very large impact on efficiency.
6)	Does the code link in the paper expose the author information? (PhD student of Tsinghua University in China, Department of Automation).


**Summary Of The Paper:**

This paper studies the problem of the heterogeneity of local data distributions in federated learning. The authors argued that the distribution overlaps are not consistent but scattered in local clients. They proposed to infer the local data manifolds to learn from the informative overlaps and estimate the data density to generate samples from the local manifold. Experimental results verified the effectiveness of the proposed method.

**Summary Of The Review:**

The paper raises relatively novel research questions. However, the proposed method has some shortcomings in motivations and privacy protection. In addition, the authors seem to have exposed personal information.
===============
I checked the author's rebuttals, some of them are useful and clarifying my concerns. I am choose to improve my scorings.

---

> ### Author Response · Authors · 2022-11-18
> **Part 1**
>
> We would like to thank the reviewer for the very insightful and valuable comments. Below are our responses to the comments.
>
> Firstly, we would like to explain a key point:
> * **the relationship between PCFL and the proposed communication-efficient algorithm using the similarity metric.**
>
>     **Answer:** (1). PCFL aims to learn models using all beneficial parts of data in other clients and excluding the effect of other ambiguous data shards, which is formulated in Eq.(14);
>
>     (2). the proposed similarity metric in Sec. 4.5 is an application of PCFL. Meanwhile, the algorithm induced by the similarity metric formulated in Eq.(16) is irrelevant to the implementation of PCFL;
>
>     (3). all experiments in the main text are conducted using our framework PCFL rather than the similarity metric;
>
>     (4). To lessen the chances for misunderstanding, we put all content about the similarity metric in Appendix in our revised submission.
>
> To the comments in ****Weaknesses:****
> * ****Question 1: In section 4.3, the authors used the normalizing flow to infer the local data manifolds. However, there is no motivation to separate the latent space into two parts. What is the point of doing so?****
>
>     ****Answer:**** We would like to explain it as follows:
>
>     (1). **a sparse representation improves the inference of the real low-dimensional manifold.** The true manifold is agnostic and low dimensional. However, a normal normalizing flow requires the latent space to have the same dimension as the original high-dimensional data. Following the work in manifold learning [1,2], we approximate the low dimensional manifold by separating the latent space for a sparse representation.
>
>     (2). **separating the latent space significantly facilitates the optimization of the normalizing flow.** As we mentioned above, NF requires the latent space to have the same dimension as the original data. When we learn the manifold of high-dimensional data, the optimization of NF could have extensive computation overhead because of high dimensional latent space. By separating the latent space into two parts, we significantly reduce the computation overhead.
>
>     [1]. Brehmer J, Cranmer K. Flows for simultaneous manifold learning and density estimation[J]. Advances in Neural Information Processing Systems, 2020, 33: 442-453.
>
>     [2]. Caterini A L, Loaiza-Ganem G, Pleiss G, et al. Rectangular flows for manifold learning[J]. Advances in Neural Information Processing Systems, 2021, 34: 30228-30241.
>
> * ****Question 2: The authors ... However, the authors did not explain how to learn the parameters of the two transformations in the method and experiment section.****
>
>     ****Answer:**** We would like to clarify that
>
>     (1). **We did introduce how to learn the parameters of the two transformations in Alg. 1 in Appendix in our original submission** because of the page limit;
>
>     (2). Following the advice, we show the learning of $g_{\theta}$ and $h_{\phi}$ in the method section in our revised submission.
> * ****Question 3: In Equation (16), the authors required client i to collaborate with certain clients who have a higher client similarity. But it is confusing to minimize the loss of the model on other clients’ datasets.****
>
>     ****Answer:**** We would like to explain it as follows:
>
>     (1). Eq.(16) formulates the proposed another communication-efficient algorithm, which is an application of PCFL formulated in Eq.(14) and it is irrelevant to the implementation of PCFL;
>
>     (2). Similar to Clustered FL [1] which minimizes the loss of the model on the clients in a cluster, the algorithm formulated in Eq. (16) learns models by minimizing the loss of the model on a group of clients with a higher similarity.
>
>     (3). To lessen the chances of misunderstanding on PCFL, we put all content about similarity metric in Appendix in our revised submission.

---

> ### Author Response · Authors · 2022-11-18
> **Part 2**
>
> * ****Question 4: In Section 4.4, the final objective in Equation (14) uses both the data of client i and part of the data of other clients to train the local model. However, this private data should be kept locally on each client. So how did the authors train models with the data at the same time?****
>
>     ****Answer:**** We would like to explain that PCFL has no additional risk of privacy leakage compared with FedAvg. The reasons are as follows:
>
>     (1). **the private data are always kept locally on each client during the training of PCFL.** From From Alg. 1 and Alg. 2 in Appendix in our original submission, the pipeline of the realization of PCFL formulated in Eq.(14) is as follows: 1. the server trains a NF model to approximate the manifold by transferring gradient information (see Alg. 1); 2. the server identifies the overlaps of the manifold without access to raw data (see Alg. 2); 3. the server trains local models by aggregating the gradients of all clients. Note that other clients only send the sum of part of the gradient;
>
>     (2). **The invertible representations are not allowed to be shared in PCFL.** In the realization of PCFL, the invertible representation is locally computed using the raw data from local clients. Throughout the whole model learning process, the invertible representation is not allowed to be shared.
>
> * ****Question 5: The authors should experiment with the sensitivity of the parameters of the proposed method, such as the threshold $\epsilon$, which has a very large impact on efficiency.****
>
>     ****Answer:**** We would like to explain it as follows:
>
>     (1). as we mentioned above, the algorithm formulated as Eq.(16) is not our main framework PCFL. PCFL is formulated in Eq.(14), and we did ablation studies about the sensitivity of the parameters of PCFL in Appendix in our original submission.
> For your easy reference, we copy the results from our original submission in the following table.
>
>  | Dataset | local | $\alpha$ = 0 | $\beta$ = 0 | PCFL |
>  | ---- | ---- | ---- | ---- | ---- |
>  | CIFAR10 | $68.9_{\pm 1.1}$ |  $76.7_{\pm .3}$ |  $79.4_{\pm .7}$ | $84.1_{\pm .8}$ |
>  | FEMNIST | $59.9_{\pm .9}$ |  $75.5_{\pm 1.1}$ |  $82.7_{\pm .8}$ | $89.7_{\pm .2}$ |
>   | CelebA | $69.3_{\pm 1.1}$ |  $82.6_{\pm 2.7}$ |  $80.8_{\pm .9}$ | $85.8_{\pm 1.1}$ |
>    | eICU | $73.7_{\pm .4}$ |  $76.6_{\pm .3}$ |  $77.4_{\pm .5}$ | $78.6_{\pm .4}$ |
>
> We set $\alpha = 0$ to evaluate the performance if we only use the generated data sampled from a learned distribution density, and set $\beta = 0$ to evaluate the performance if we only use the identified overlaps from other clients.
>
> (2). the algorithm in Eq.(16) is just an extra application of our framework for overlap identification. Following the advice, we experiment with the sensitivity of $\epsilon$. The $\epsilon$ parameter could be adaptive in our implementation. We set $\epsilon$ by controlling the number of clients to collaborate for each client. There are 14 clients in eICU dataset. We test collaborating client numbers to be 1, 3, 5, etc. The results are shown in the following table.
>
> | | | | | | | | |
> |:----:|:----:|:----:|:----:|:----:|:----:|:----:|:----:|
> |****client number**** | 1 | 3 | 5 | 7 | 9 | 11 | 13 |
> |****AUC (%)****       | $69.0_{\pm .4}$ | $75.4_{\pm .2}$ | $75.5_{\pm .7}$ | $78.0_{\pm .1}$ | $77.1_{\pm .9}$ | $76.8_{\pm .2}$ | $77.0_{\pm .3}$ |
> | | | | | | | | |
>
> * ****Question 6: Does the code link in the paper expose the author information?****
>
>     ****Answer:**** We would like to clarify it as follows:
>
>     (1). **after a thorough examination of our code link, we cannot find any exposed author information.** We have checked all uploaded files (including the source codes, README.md, the referred links, etc.), but did not find any information relevant to the author/affiliation;
>
>     (2). **a public GitHub repository may be unintentionally modified/edited by any third party.** We guess what you mentioned is another GitHub account who has watched/edited our repository by accident. (If it is not the case, please feel free to point it out at any time.)
>
>     (3). **we try our best to set the GitHub repository to keep as much information confidential.** Following the advice, we have placed restrictions on our code repository and now it will not be modified by any third party.

---

> > ### Comment · Reviewer_XA4j · 2022-11-25
> > **Some questions about the answers to Question 4**
> >
> > Thank you for answering my doubts, but I still have some confusion. Is the gradient for each client calculated based on all its data? Or is the gradient calculated only on the basis of partially favorable data? How to train a local client model using only part of the data of other clients?

---

> > > ### Author Response · Authors · 2022-11-25
> > > **Reply to Reviewer XA4j**
> > >
> > > We would like to explain it as follows.
> > >
> > > **The gradient is calculated only on the basis of partially favorable data.**
> > >
> > > We would like to explain our training pipeline as follows:
> > >
> > > 1. **for each client $D^{i}$, we identify the favorable data $D^{i,k}$ in other clients;** we identify the beneficial data overlaps by manifold learning, which determines the favorable data overlaps by computing the overlaps between local manifolds without privacy preserving;
> > >
> > > 2. **for the personalized model $f^{i}$, other clients only calculate the gradient on the identified data $D^{i,k}$.** When learning the model $f^{i}$ for the $i$-th client, other clients only use the identified subset $D^{i,k}$ to train the model $f^{i}$. Then the sum of the gradient on the basis of partially favorable data is sent to the server, as formulated in Eq. (14) in the main text.

---

> ### Author Response · Authors · 2022-11-23
> **Need Further Clarification?**
>
> Thanks very much for your constructive comments on the motivation and the privacy of our work. We have tried our best to address the concerns. Is there any unclear point so that we should/could further clarify?

---

> ### Author Response · Authors · 2022-11-25
> **Thanks for your reply!**
>
> Many thanks for your reply!
>
> We would like to appreciate your valuable comments on the shortcomings in motivations and privacy protection. We are glad to know that our clarification alleviates these concerns and you choose to improve the score.
>
> If there are any further issues about precision collaboration, we are very happy to share our thoughts and ideas at any time.

---

### Official Review · Reviewer_kbbP · 2022-10-23

**Confidence:** 4
**Correctness:** 2
**Technical Novelty And Significance:** 3
**Empirical Novelty And Significance:** 2
**Recommendation:** 5

**Clarity, Quality, Novelty And Reproducibility:**

Clarity and Reproducability:

* The main contributions listed in Section 1 are too vague. Pls point out the theoretical
result (Prop., Them.) or methodological result (Algorithm) that you consider the main contribution.
In my opinion, the main contribution of the paper is the application of basic manifold learning
techniques to obtain a measure for the similarity between two local dataset. If this is the case,
then authors should explain what the pro and con of this new measure is compared e..g, to KL divergence
of Wasserstein distance.

* The numerical experiments needs to be explained in more detail. At least Section 5.1. and 5.2.
lack many details on how the proposed PCFL method is applied. What are the hypothesis spaces used
to fit the local datasets? What is the dimension d' of the latent space used for the manifold learning step?
Also, how did you precisely compute the accuracies listed in Tables 1 -3 ? are these computed on a val. or
test set ? How did you tune the hyper-parameters of the baseline methods ?


Novelty:

* The novelty of the approach via manifold learning should be motivated better. What is the specific
challenge in using manifold learning as a proxy for distribution learning required to find out which
local datasets to pool ? What is the fundamental new idea that sets your approach apart from existing
work on clustered FL such as Ghosh et al. (2020)? I do not understand what you mean by "However, their
hypothesis excludes the possibility of knowledge transfer across clusters."

* Authors should also compare their approach to graph-based methods for federated learning.
These methods use a similarity graph, representing pair-wise similarities of distributions
underlying local datasets, to construct regularisation terms. A popular choice for this
regularization term are measures of total variation as used by network Lasso

Network Lasso: Clustering and Optimization in Large Graphs
D. Hallac, J. Leskovec, and S. Boyd, Proceedings SIGKDD, pages 387-396, 2015.

These total variation based methods for FL provide a smooth trade-off between learning tailored local
models (that might overfit if local data is too small) and a common global model (that might under-fit
for heterogeneous local datasets). This trade-off has been studied in a recent line of work:

A. Jung, "Networked Exponential Families for Big Data Over Networks," in IEEE Access, vol. 8, pp. 202897-202909, 2020, doi: 10.1109/ACCESS.2020.3033817.

A. Jung and N. Tran, "Localized Linear Regression in Networked Data," in IEEE Signal Processing Letters, vol. 26, no. 7, pp. 1090-1094, July 2019, doi: 10.1109/LSP.2019.2918933.

The two latter works study the relation between local models (their loss function), similarity
structure between local datasets and the resulting cluster structure. This analysis might be combined
with your similarity measure to obtain theoretical performance bounds for your proposed method. Moreover,
it would be interesting to compare these similarity graph based methods with your approach. It seems
that you can use your similarity measure to obtain the edges (and their weights) of the similarity/empirical
graph used by these methods.

Quality:

* pls explain the notation used for the expectations in (14)

* as to the first contribution: what do you mean precisely by "..shared knowledge is fragmented among local clients;" ?

* as to the second contribution: how do you show that your method achieves a more precise collaboration ?

* as to the third contribution: "Because of the adaptability of our proposed framework.." pls explain in more detail
how your method offers the claimed adaptability.


* what is a “federated network” ?

* what precisely is an "overlap"  ? is it a subset/region of the sample space?

* "While it is hard to learn the local manifold from the insufficient data in local clients directly,"
why and in what precise sense is the local client data insufficient ?

* i do not understand the meaning of Eq. (2)

* "While the data manifold of local clients is mostly agnostic..." what is an agnostic data manifold ?

* "We suggest leveraging the overlaps via the learned data manifold to prevent privacy leakage." Do you study the privacy leakage of your method in more detail ?

* it is unclear how PCFL is combined with "baseline" methods to obtain the results depicted in Figure 7, 8

* "However, i.i.d. assumption in Eq.(2) is largely violated as the local data distributions may be
significantly distinctive." pls justify this claim with citations or describing a specific application domain where
i.i.d. assumption is violated.

* “We intuitively show the motivation of our method by conducting experiments on synthetic data. “
Im not sure if numerical experiments are effective for building intuition about some method.

* "..needs to precisely identify the data overlap sampled from.." this is unclear to me.

**Strength And Weaknesses:**

Strength:
The manifold learning perspective to FL from heterogeneous data is novel to the best of my knowledge. Authors do a good job in motivating and explaining this perspective on a high level.

Weaknesses:
* The novelty and relevance of the proposed approach needs more motivation and explanation.
* More theoretical analysis of the proposed method (generalization bounds, computational complexity) is required.
* The numerical experiments should include comparison with clustered FL and graph-based FL methods such as network Lasso.




**Summary Of The Paper:**

The papers approaches federated learning from heterogeneous data from a manifold learning perspective. Each local datasets is obtained by sampling from a local manifold. The intersections or overlaps of these local manifolds is used to measure the similarity between local datasets.

**Summary Of The Review:**

see above.

---

> ### Author Response · Authors · 2022-11-18
> **Part 1**
>
> We would like to thank the reviewer for the very insightful and valuable comments. Firstly, we would like to clarify several key points.
> * **the relationship between PCFL and the proposed communication-efficient algorithm using the similarity metric.**
>
>     **Answer:** (1). PCFL aims to learn models using all beneficial parts of data in other clients and excluding the effect of other ambiguous data shards, which is formulated in Eq.(14);
>
>     (2). the proposed similarity metric in Sec. 4.5 is an application of PCFL. Meanwhile, the algorithm induced by the similarity metric formulated in Eq.(16) is irrelevant to the implementation of PCFL;
>
>     (3). all experiments in the main text are conducted using our framework PCFL rather than the similarity metric;
>
>     (4). To lessen the chances for misunderstanding, we put all content about the similarity metric in Appendix in our revised submission.
>
> * **The difference between PCFL and Clustered FL.**
>
>     **Answer:** (1). **clustered structure hypothesis and fragmented information sharing:** clustered FL assumes that the local clients are naturally partitioned into several clustered; PCFL assumes that there may be no clustered structure but fragmented data in different clients share common distributions.
>
>     (2). **global model and personalized model:** clustered FL learns a common model for all clients in the cluster; PCFL learns a personalized model for each client;
>
>     (3). **clustered collaboration and precision collaboration:** clustered FL learns models by minimizing the loss on all data in the cluster; PCFL learns personalized models by minimizing the loss on the identified beneficial data parts in all other clients.
>
> Below are our responses to the comments.
>
> To the comments in ****Weaknesses:****
> * ****Question 1: The novelty and relevance of the proposed approach need more motivation and explanation.****
>
>     ****Answer:**** We would like to explain it briefly as follows (a more detailed explanation could be found in the following):
>
>     (1). **We investigate a more general and practical FL scenario.** Previous FL research assumes that there exists a common representation or clustered structure. PCFL argues that the real data could be very diverse and the above hypothesizes are largely violated. Fragmented information sharing is more practical in reality;
>
>     (2). **Compared with baselines, PCFL achieves a "precise" collaboration for each client.** Existing methods learn models using the whole dataset from other clients. However, there may be harmful data in other clients. Precise identification of the beneficial overlaps could facilitate the model learning;
>
>     (3). **PCFL achieves an efficient collaboration in a privacy-preserving way.** Identifying the shared overlaps between each pair of clients could have $O(N^{2})$ time complexity. We identify the overlaps by approximating the low dimensional latent manifold, which avoids possible privacy leakage and significantly reduces the computation overhead.
>
> * ****Question 2: More theoretical analysis of the proposed method (generalization bounds, computational complexity) is required.****
>
>     ****Answer:**** We would like to thank you for the kind advice. Following the advice, we discuss the computational complexity in Appendix in our revised submission.
>
> * ****Question 3: The numerical experiments should include comparison with clustered FL and graph-based FL methods such as network Lasso.****
>
>     ****Answer:**** (1). Following the suggestion, we show the results using the clustered FL in our experiments. Since the network Lasso cannot be used in federated learning scenarios directly, we implement a graph-based FL method and show the results in the experiments in our revised submission.
>
>     (2). Clustered FL or graph-based FL learn models using all data of the clients in the cluster, which could result in inefficient data utilization, because there may be harmful data in the cluster and beneficial data outside the cluster. From Table 6 in Appendix, PCFL still outperforms all baselines.

---

> ### Author Response · Authors · 2022-11-18
> **Part 2**
>
> To the comments in ****Clarity, Quality, Novelty And Reproducibility:****
>
> * ****Question 4: The main contributions listed in Section 1 are too vague.****
>
>     ****Answer:**** We would like to clarify it as follows:
>
>     (1). **our main contribution is the precise identification of the beneficial overlaps between local clients.** Existing methods cannot achieve precise collaboration but a coarse data utilization from other clients. For example, clustered FL aims to measure the similarity between clients and learn a global model for each cluster. However, such a global model may not be an optimal model for each client in the cluster. We propose to learn a model for each client using the data overlaps from all other clients.
>
>     (2). **the similarity measurement is just an application of PCFL, which is put in Appendix in our revised submission.** Our method for the identification of the overlaps could be used for similarity measurement. So we propose another communication-efficient algorithm with the similarity metric in Sec. 4.5. We also experimentally verify it in Sec. D.1 in Appendix in our original submission.
>
>     (3). **The referred similarity measurements (KL divergence or Wasserstein distance) are is forbidden in federated learning because of privacy concerns.** The calculating of Wasserstein distance requires direct access to the raw data of local clients, which could raise privacy concerns. In the realization of our proposed precise collaboration, we identify the overlaps by the approximated local manifold, which avoids potential privacy leakage.
>
>     (4). Following the advice, we try our best to revise Sec. 1 in our paper, which may be more clear about the summarization of our main contribution.
>
>
> * ****Question 5: The numerical experiments need to be explained in more detail.****
>
>     ****Answer:**** In our original submission, we added more details about the numerical experiments in Appendix. We would like to explain the referred questions as follows.
>
>     **Question 5.1: What are the hypothesis spaces used to fit the local datasets?**
>
>     **Answer:** The hypothesis space $\mathcal{F}$ defined in Eq.(16) denotes all possible deep models. In the experiments, we follow the work [1] to build the network structure to determine the hypothesis space.
>
>     **Question 5.2: What is the dimension d' of the latent space used for the manifold learning step?**
>
>     **Answer:** **d' denotes the dimension of the low-dimensional manifold.** A classical NF requires the latent variable $e \in E$ to have the same dimension with the data space $Z$. We separate the latent space $E = U \ \times \ V$, where $U = \mathbb{R}^{d'}$ denotes the coordinates on the manifold. $V = \mathbb{0}^{d-d'}$ denotes the remaining coordinates, which are the directions orthogonal to the manifold.
>
>     **Question 5.3: how did you precisely compute the accuracies listed in Tables 1-3?**
>
>     **Answer:** Following the convention [1], we first compute the accuracies of test sets in each client. Then we calculate the weighted average of these accuracies according to the number of test samples in each client.
>
>     **Question 5.4: are these computed on a val. or test set?**
>
>     **Answer:** Following baselines [1], all experimental results are computed on the test set.
>
>     **Question 5.5: How did you tune the hyper-parameters of the baseline methods?**
>
>     **Answer:** Following baselines [1], we apply the same setting (including training epoch, batch size, federated setting, etc.) in our method. The hyper-parameters are determined by comparing the performance on the training set [1].
>
> [1]. Liam Collins, Hamed Hassani, Aryan Mokhtari, and Sanjay Shakkottai. Exploiting shared representations for personalized federated learning. 2021.
>
> To the comment in **Novelty:**
> * **Question 6: The novelty of the approach via manifold learning should be motivated better.**
>
>     **Answer:** We would like to explain our motivation for manifold learning as follows:
>
>     (1). **with the manifold learning, we identify the beneficial data parts in other clients.** Existing methods learn models using the whole data of other clients, which could hurt the model learning when there are harmful data;
>
>     (2). **with the manifold learning, our approach achieves an efficient model learning in a privacy-preserving way.** We approximate the low dimensional manifold to identify the overlaps, which avoids direct access to the raw data and improves the efficiency of the optimization.

---

> ### Author Response · Authors · 2022-11-18
> **Part 3**
>
> * **Question 7: What is the specific challenge in using manifold learning as a proxy for distribution learning required to find out which local datasets to pool?**
>
>     **Answer:** We would like to explain it as follows:
>
>     (1). **privacy concern:** existing distribution learning methods (e.g., KL divergence, Wasserstein distance) may require to get access to the raw data of the clients, which could raise privacy concerns. We propose to use manifold learning to find out the beneficial subsets (overlaps) without access to the raw data, which protects private information;
>
>     (2). **optimization:** learning the manifold directly using a normalizing flow method could have a large computation overhead. PCFL learns a low-dimensional manifold by separating the latent space, which improves the optimization efficiency in high-dimensional scenarios.
>
> * **Question 8: What is the fundamental new idea that sets your approach apart from existing work on clustered FL such as Ghosh et al. (2020)?**
>     **Answer:** We would like to explain it as follows:
>
>     (1).**Clustered FL could not distinguish beneficial/harmful parts in a local dataset, but PCFL achieves it.** Clustered FL learns models using all data in the cluster, which ignores the possibly useful parts outside the cluster. More importantly, Clutered FL cannot exclude the adverse effect caused by the harmful parts in the cluster.
>
>     (2). **We experimentally verify that PCFL could achieve better model performance than Clustered FL.**
> The clustered FL algorithm estimates the cluster identities for each client. Each cluster shares a common prediction model and the parameters are optimized via average gradient descent. We conduct experiments on the eICU dataset. As shown in the table below, Clustered FL achieves comparable results as other baselines, while PCFL outperforms all the baselines.
>
> | method | AUC (%) |
> | ---- | ---- |
> | local | $73.7_{\pm 1.4}$ |
> | FedAvg | $73.2_{\pm .5}$ |
> | FedProx | $78.2_{\pm .2}$ |
> | Fed-MTL | $77.2_{\pm 1.6}$ |
> | PerFedAvg | $73.8_{\pm .3}$ |
> | LG-FedAvg | $74.5_{\pm .2}$ |
> | FedPer | $74.3_{\pm .7}$ |
> | FedRep | $74.1_{\pm 1.2}$ |
> | APFL | $68.3_{\pm .8}$ |
> | L2GD | $72.0_{\pm .6}$ |
> | Ditto | $78.3_{\pm .1}$ |
> | ****Clustered FL**** | $74.7_{\pm .3}$ |
> | ****PCFL**** | $80.0_{\pm .6}$ |
>
> * **Question 9: I do not understand what you mean by "However, their hypothesis excludes the possibility of knowledge transfer across clusters.**
>
>     **Answer :** We would like to explain that the meaning is:
>
>     (1). Clustered FL learns models using the data within the cluster. However, there may be favorable data shards outside the cluster, but Clustered FL cannot utilize these data to facilitate the model learning.
>
>     (2). Clustered FL assumes that the local datasets are partitioned into different clusters. It may be a little harsh in reality. Therefore, the hypothesis excludes the possibility of knowledge transfer across clusters [1].
> [1]. Marfoq O, Neglia G, Bellet A, et al. Federated multi-task learning under a mixture of distributions[J]. Advances in Neural Information Processing Systems, 2021, 34: 15434-15447.
>
> * ****Question 10: Authors should also compare their approach to graph-based methods for federated learning. These methods use a similarity graph, representing pair-wise similarities of distributions underlying local datasets, to construct regularisation terms. A popular choice for this regularization term is measures of total variation as used by network Lasso."****
>
>     ****Answer:**** We thank you for the kind advice on the comparisons with graph-based methods. We would like to explain it as follows:
>
>     (1). **graph-based methods cannot achieve "precision" collaboration.** Similar to clustered FL, graph-based methods learn models using all data in the cluster, which cannot fully utilize all beneficial data overlaps outside the cluster and could be hurt by the harmful data in the cluster.
>
>     (2). **Experiments verify that PCFL outperforms graph-based baselines.** Following the advice, we conduct experiments of Network Lasso (Hallac, 2015) on eICU dataset. Network Lasso allows for simultaneous clustering and optimization on graphs. However, it couldn't be directly applied to federated learning due to privacy issues. We use average sample distance as edge weight between clients and optimize by gradient descent, so privacy is preserved by transmitting aggregated gradients. As the results displayed below, PCFL achieves better performance than Network Lasso.
>
> | method | AUC (%) |
> | ---- | ---- |
> | local | $73.7_{\pm 1.4}$ |
> | FedAvg | $73.2_{\pm .5}$ |
> | FedProx | $78.2_{\pm .2}$ |
> | Fed-MTL | $77.2_{\pm 1.6}$ |
> | PerFedAvg | $73.8_{\pm .3}$ |
> | LG-FedAvg | $74.5_{\pm .2}$ |
> | FedPer | $74.3_{\pm .7}$ |
> | FedRep | $74.1_{\pm 1.2}$ |
> | APFL | $68.3_{\pm .8}$ |
> | L2GD | $72.0_{\pm .6}$ |
> | Ditto | $78.3_{\pm .1}$ |
> | ****Network Lasso**** | $76.3_{\pm .8}$ |
> | ****PCFL**** | $80.0_{\pm .6}$ |

---

> ### Author Response · Authors · 2022-11-18
> **Part 4**
>
> * ****Question 11: The two latter works study the relation between local models (their loss function), the similarity structure between local datasets, and the resulting cluster structure. This analysis might be combined with your similarity measure to obtain theoretical performance bounds for your proposed method."****
>
>     ****Answer:**** We would like to explain it as follows:
>
>     (1). **PCFL is not used to obtain the cluster structure, but identifies all beneficial data overlaps.** Different from clustered FL, PCFL formulated in Eq.(14) learns models using the data overlaps in all local datasets, rather than the datasets in the cluster;
>
>     (2). **Similarity measure is just an application of PCFL.** To lessen the chances for misunderstanding, we put all content about the similarity metric in Appendix in our revised submission;
>
>     (3). Following the advice, we cite and discuss the referred two works in our revised submission. , we analyze the computation complexity of our proposed method in our revised submission.
>
> * ****Question 12: Moreover, it would be interesting to compare these similarity graph based methods with your approach. It seems that you can use your similarity measure to obtain the edges (and their weights) of the similarity/empirical graph used by these methods."****
>
>     **Answer: ** We thank you for the advice on the similarity measure. Following the advice, we discuss the application of the referred methods in our proposed similarity measure in our revised submission.
>
> To the comments in ****Quality:****
>
> * ****Question 13: pls explain the notation used for the expectations in (14)."****
>
>     ****Answer:**** Eq.(14) formulates our proposed framework PCFL.
>
>     (1). $\left(x^{k}, y^{k}\right) \in D^{i, k}$ denotes the data points from the identified data overlaps between the $i$-th client and $k$-th client;
>
>     (2). $(x, y) \sim p_{z}^{i}(z)$ means the data points are sampled from the learned manifold based on the approximated distribution density $p_{z}^{i}(z)$.
>
> * ****Question 14: as to the first contribution: what do you mean precisely by "..shared knowledge is fragmented among local clients;"?"****
>
>     ****Answer:**** The meaning is that there may be no two datasets i.i.d. The shared knowledge is implied in the shared data. The shared data (the data in different clients satisfying i.i.d. assumption) are fragmented.
>
> * ****Question 15: as to the second contribution: how do you show that your method achieves a more precise collaboration?"****
>
>     ****Answer:**** We would like to explain it as follows:
>
>     (1). **direct demonstration:** the true manifold of the raw data is mostly agnostic. In our numerical experiments, PCFL effectively learns the manifold and identifies the overlapped data shown in Figure 5(e) (the points with the same color denote the overlapped data.).
>
>     (2). **indirect demonstration:** a more precise collaboration means a better model performance. From the experiments shown in Sec. 5.2 and Sec. 5.3, PCFL significantly outperforms all baselines.
>
> * ****Question 16: as to the third contribution: "Because of the adaptability of our proposed framework.." pls explain in more detail how your method offers the claimed adaptability."****
>
>     ****Answer:**** We would like to explain it as follows:
>
>     (1). **The adaptability means that PCFL formulated in Eq.(14) could be used in other FL algorithms.** From Eq.(14), PCFL precisely uses the shared data overlap in other clients (the second term in Eq.(14)), and uses the generated data sampled from the distribution density (the third term in Eq.(14));
>
>     (2). **PCFL significantly improves the performance of baselines.** Since the second and third terms in Eq.(14) could be used in other baselines, we experimentally validate the performance of PCFL+baseline. The results in Sec. 5 show that PCFL significantly improves the performance of baselines.
>
>
> * ****Question 17: what is a “federated network”?"****
>
>     ****Answer:**** Federated network in related works [1,2,3] usually denotes a federated learning system where there are clients seeking to learn models collaboratively.
>
>     [1]. Li T, Sahu A K, Zaheer M, et al. Federated optimization in heterogeneous networks[J]. Proceedings of Machine Learning and Systems, 2020, 2: 429-450.
>
>     [2]. Li T, Sanjabi M, Beirami A, et al. Fair Resource Allocation in Federated Learning[C]//International Conference on Learning Representations. 2019.
>
>     [3]. Li T, Hu S, Beirami A, et al. Ditto: Fair and robust federated learning through personalization[C]//International Conference on Machine Learning. PMLR, 2021: 6357-6368.

---

> ### Author Response · Authors · 2022-11-18
> **Part 5**
>
> * ****Question 18: what precisely is an "overlap"? is it a subset/region of the sample space?****
>
>     ****Answer :**** Yes. The distribution overlap means that the data of two clients have overlapped patterns. The overlapped data are identified as the subsets of the local datasets that are independently identically distributed (i.i.d).
>
> * ****Question 19: "While it is hard to learn the local manifold from the insufficient data in local clients directly," why and in what precise sense is the local client data insufficient?****
>
>     ****Answer:**** We would like to explain that:
>
>     (1). **Local client data is insufficient in FL scenarios.** In FL, since local clients cannot learn a promising model by local training because of insufficient data, they choose to collaborate with other clients;
>
>     (2). **Learning local manifolds needs more data than learning models for classification.** A generative model needs more data samples than discriminative learning [1]. Therefore, in this sense, it is hard to learn the local manifold from insufficient data.
>
>     [1]. Song Y, Ermon S. Generative modeling by estimating gradients of the data distribution[J]. Advances in Neural Information Processing Systems, 2019, 32.
>
> * ****Question 20: i do not understand the meaning of Eq. (2).****
>
>     ****Answer:**** Eq.(2) means that for each dataset $D^{i} \ (0 \leq i \leq N-1)$:
>
>     (1). $D^{i}$ is sampled from a common manifold $D^{i} \subset \mathcal{M}$;
>
>     (2). $D^{i}$ is sampled with a common distribution density (the local datasets are i.i.d.), which is denoted as $z \in D^{i}, z \sim p^{g}_{z}$.
>
> * ****Question 21: "While the data manifold of local clients is mostly agnostic..." what is an agnostic data manifold?****
>
>     ****Answer:**** In reality, the true data manifold is mostly agnostic. For example, suppose $\mathcal{M}$ denotes the manifold of the image of an animal, we cannot obtain the exact $\mathcal{M}$ but approximate $\mathcal{M}$.
>
> * ****Question 22: "We suggest leveraging the overlaps via the learned data manifold to prevent privacy leakage." Do you study the privacy leakage of your method in more detail?****
>
>     ****Answer: **** We would like to explain it as follows:
>
>     (1). **identifying data overlaps directly could cause privacy leakage.** Identifying data overlaps directly requires to have access to the raw data of different clients simultaneously, which could cause privacy leakage;
>
>     (2). **learning manifolds for identifying overlaps avoid getting access to the raw data directly.** PCFL learns models by transferring gradient information as FedAvg does, which has no additional risk of privacy leakage.
>
>
> * ****Question 23: it is unclear how PCFL is combined with "baseline" methods to obtain the results depicted in Figure 7, 8.****
>
>     ****Answer: **** We would like to explain it:
>
>     (1). **the identified distributional overlaps and the generated data via the learned manifold could be directly applied to other methods.** From Eq.(14), the second and the third terms, which denote the identified distributional overlaps and the generated data via the learned manifold respectively, could be added to the objectives of baselines;
>
>     (2). **PCFL+baseline is formulated by combining the original baseline objective, the second term in Eq.(14), and the third term in Eq.(14) together.** The results depicted in Figure 6-10 in our original submission demonstrate the effectiveness of our proposed method.
>
> * ****Question 24: "However, i.i.d. assumption in Eq.(2) is largely violated as the local data distributions may be significantly distinctive." pls justify this claim with citations or describing a specific application domain where i.i.d. assumption is violated.****
>
>     ****Answer:**** We would like to explain it as follows:
>
>     (1). **justify this claim with citations:** Previous works in FL about heterogeneity have verified this claim, for example [1,2];
>
>     (2). **a specific application domain where i.i.d. assumption is violated.** A real clinical research network (CRN)[3], has multiple hospitals located in different regions. Because of different patient populations, there is research showing that there is statistical heterogeneity in CRN, where the i.i.d. assumption is seriously violated.
>
>     [1]. Mehryar Mohri, Gary Sivek, and Ananda Theertha Suresh. Agnostic federated learning. In International Conference on Machine Learning, pp. 4615–4625. PMLR, 2019.
>
>     [2]. Liam Collins, Hamed Hassani, Aryan Mokhtari, and Sanjay Shakkottai. Exploiting shared representations for personalized federated learning. In ICML 2021.
>
>     [3]. Rachael L Fleurence, Lesley H Curtis, Robert M Califf, Richard Platt, Joe V Selby, and Jeffrey S Brown. 2014. Launching PCORnet, a national patient-centered clinical research network. Journal of the American Medical Informatics Association 21, 4 (2014), 578–582.

---

> ### Author Response · Authors · 2022-11-18
> **Part 6**
>
> * ****Question 25: “We intuitively show the motivation of our method by conducting experiments on synthetic data. “ I'm not sure if numerical experiments are effective for building intuition about some method.****
>
>     ****Answer:**** We would like to explain it as follows:
>
>     (1). **We can only use synthetic data to judge whether PCFL obtains the manifold.** The true manifold of the real data is mostly agnostic. To evaluate the learned manifold, researchers choose to conduct numerical experiments where the manifold is pre-defined;
>
>     (2). **Experimental verification:** From Sec. 5.1, PCFL learns a promising manifold shown in Fig. 5(e), and identifies the overlapped data in local datasets. From Fig. 5(a-e), PCFL learns a better model compared with baselines.
>
> * ****Question 26: "..needs to precisely identify the data overlap sampled from.." this is unclear to me.****
>
>     ****Answer:**** We would like to explain it as follows.
>
>     (1). The learning task on the client $D^{0}$ is the regression of $y$ using $x$ sampled from the range [0, $\frac{\pi}{2}$] and [$\frac{\pi}{2}$, $\pi$], while the learning task on the client $D^{1}$ is the regression of $y$ using $x$ sampled from the range [$\frac{\pi}{2}$, $\pi$] and [$\pi$, $\frac{3\pi}{2}$];
>
>     (2). For $D^{0}$, the best way to collaborate with $D^{1}$ is precisely use the data sampled from [$\frac{\pi}{2}$, $\pi$] in $D^{1}$, because [$\frac{\pi}{2}$, $\pi$] is the distribution overlaps between the two clients. Therefore, we say that $D^{0}$ needs to precisely identify the data overlap sampled from [$\frac{\pi}{2}$, $\pi$] in $D^{1}$.

---

> ### Author Response · Authors · 2022-11-25
> **Need further clarification?**
>
> Thanks very much for your constructive comments on the novelty and quality of our work. We have tried our best to address the concerns. Is there any unclear point so that we should/could further clarify?

---

### Official Review · Reviewer_psQN · 2022-10-23

**Confidence:** 3
**Correctness:** 3
**Technical Novelty And Significance:** 3
**Empirical Novelty And Significance:** 3
**Recommendation:** 5

**Clarity, Quality, Novelty And Reproducibility:**

The paper is relatively clear. However, the role of certain equation and values of hyperparamters are not immediately obvious from the main paper.

**Details Of Ethics Concerns:**

The paper heavily relies on the reconstruction of local data using an *invertible* generative model. Even though the paper claims to preserve privacy by not sharing clients’ data directly, invertible hidden representations of local data are shared from the clients to the server and among clients. The performance improvement might have come from the high-fidelity reconstruction of the original image from the hidden representations using the NL model. This could defeat the purpose of privacy preservation, especially when a powerful invertible generative model is involved [1].

[1] Hitaj, Briland, Giuseppe Ateniese, and Fernando Perez-Cruz. "Deep models under the GAN: information leakage from collaborative deep learning." Proceedings of the 2017 ACM SIGSAC conference on computer and communications security. 2017.



**Strength And Weaknesses:**

Pros:

1, The paper cleverly uses the properties of normalizing flow, e.g., exact likelihood and invertibility, to find data overlaps and sample additional training data.

2, The proposed algorithm demonstrates performance  on multiple benchmarks.


Cons:

**1, Privacy concern on sharing invertible representations.** Even though the paper claims to preserve privacy by not sharing clients’ data directly, invertible hidden representations of local data are shared from the clients to the server and among clients. The performance improvement might have come from the high-fidelity reconstruction of the original image from the hidden representations using the NL model.  This could defeat the purpose of privacy preservation, especially when a powerful invertible generative model is involved [1].

**2, Contribution of each of the components not clear.** Two types of generative data are used on each client: inverted data from other clients and sampled data from the learned local manifold. Only the first one is directly related to collaboration among clients. It is not clear how much improvement is from collaboration and how much is from the generated data. It is also not clear how many generated data from the local manifold is used, which could be an important factor to the improvement. An ablation study would resolve this concern.


Minor:

**3, Clarity on the use of the similarity metric.** It is not clear how eq.16 is used in the final algorithm and how the hyperparameter $\epsilon$ is set.

[1] Hitaj, Briland, Giuseppe Ateniese, and Fernando Perez-Cruz. "Deep models under the GAN: information leakage from collaborative deep learning." Proceedings of the 2017 ACM SIGSAC conference on computer and communications security. 2017.

**Summary Of The Paper:**

The paper proposes to use normalizing flow (NF), an invertible generative model, to improve performance of personalized FL under a fragmented sharing setting, where local data distributions partially overlap. The overall goal is to identify clients with shared data and utilize information from those clients to boost performance. Specifically, a global data manifold is learned by the generative model. The learned data manifold is then used to construct local data manifolds, which are split into shared and unique components. For the shared component, the algorithm collects invertible representations of the shared data from the other clients and invert them back to the image space using NF. For the unique component, the algorithm samples additional generated data from the learned local data manifold.

**Summary Of The Review:**

The paper presents a novel method for personalized FL. However, the reviewer has severe concerns over privacy issues raised in the method. Specifically, invertible hidden representations of local data are passed from the clients to the server and shared among clients. If the author could comment on this issue, I will be happy to raise my score.

---

> ### Author Response · Authors · 2022-11-18
> **Response to Reviewer psQN**
>
> We would like to thank the reviewer for appreciating our novelty and the very valuable comments. Below are our responses to the comments.
>
> Firstly, we would like to explain a key point:
> * **the relationship between our framework PCFL in Eq.(14) and the proposed communication-efficient algorithm using the similarity metric in Eq.(16).**
>
>     **Answer:** (1). PCFL aims to learn models using all beneficial parts of data in other clients and excluding the effect of other ambiguous data shards, which is formulated in Eq.(14);
>
>     (2). the proposed similarity metric in Sec. 4.5 is an application of PCFL. Meanwhile, the algorithm in Eq.(16) induced by the similarity metric is irrelevant to the implementation of PCFL;
>
>     (3). all experiments in the main text are conducted using our framework PCFL rather than the similarity metric;
>
>     (4). To lessen the chances for misunderstanding, we put all content about the similarity metric in Appendix in our revised submission.
>
> To the comments in ****Weaknesses:****
> * ****Question 1: Privacy concern on sharing invertible representations.****
>
>     ****Answer:**** we would like to clarify it as follows:
>
>     (1). **In our framework, the invertible representations are not allowed to be shared.** Every client cannot get access to the invertible representations of other clients. In the realization of PCFL, the invertible representation $U'^{i}$ is locally computed using the raw data in local clients. Throughout the whole model learning process, only the convex hull of the projected representation is sent to the server, which has no information about the raw data and the invertible representations. The invertible representation is not allowed to be shared.
>
>     (2). **only the gradient information is transferred.** From Alg.1 and Alg. 2, during the training of PCFL (including the manifold learning and the model learning), only the statistical information (e.g., the sum of the gradient) is transferred between the server and the clients. Therefore, PCFL has no additional risk of privacy leakage compared with FedAvg.
>
> * ****Question 2: Contribution of each of the components is not clear.****
>
>     ****Answer:**** In our original submission, we did the analysis of the contribution of each component in Sec. C in Appendix.
>
>     We copy the results from our original submission for your easy reference.
>
>  | Dataset | local | $\alpha$ = 0 | $\beta$ = 0 | PCFL |
>  | ---- | ---- | ---- | ---- | ---- |
>  | CIFAR10 | $68.9_{\pm 1.1}$ |  $76.7_{\pm .3}$ |  $79.4_{\pm .7}$ | $84.1_{\pm .8}$ |
>  | FEMNIST | $59.9_{\pm .9}$ |  $75.5_{\pm 1.1}$ |  $82.7_{\pm .8}$ | $89.7_{\pm .2}$ |
>   | CelebA | $69.3_{\pm 1.1}$ |  $82.6_{\pm 2.7}$ |  $80.8_{\pm .9}$ | $85.8_{\pm 1.1}$ |
>    | eICU | $73.7_{\pm .4}$ |  $76.6_{\pm .3}$ |  $77.4_{\pm .5}$ | $78.6_{\pm .4}$ |
>
> We set $\alpha = 0$ to evaluate the performance if we only use the generated data sampled from a learned distribution density, and set $\beta = 0$ to evaluate the performance if we only use the identified overlaps from other clients.
>
> From the above table, we have the following findings:
>
> (1). **every component effectively improves the model learning.** Compared with **local**, both $\alpha = 0$ and $\beta = 0$ achieve a better model performance;
>
> (2). **the identified overlaps could achieve more performance gain.** From the table, $\beta = 0$ performs better, which means that the shared overlaps from other clients facilitate the model learning significantly, compared with the generated data from the learned manifold.
>
> To the comments in ****Clarity on the use of the similarity metric:****
>
> * ****Question 3: It is not clear how Eq.(16) is used in the final algorithm.****
>
>     ****Answer:**** We would like to explain that:
>
>     (1). **As we stated above, Eq.(16) is an application of PCFL, which is irrelevant to our framework PCFL.** The final algorithm of PCFL is implemented according to Eq.(14), and is presented in Alg. 1 and Alg.2 in Appendix in our original submission;
>
>     (2). Since the identified overlaps could indicate the similarity between clients, we propose the communication-efficient algorithm in Eq.(16);
>
>     (3). To lessen the chances for misunderstanding, we put all content about the similarity metric in Appendix in our revised submission.
>
>
> * ****Question 4: It is not clear how the hyperparameter $\epsilon$ is set.****
>
>     **Answer:** We did experiments to evaluate the proposed communication-efficient algorithm and introduced our method about how $\epsilon$ is set in Appendix in our original submission. In particular, we set a specific $\epsilon$ for each client $D^{i}$, which satisfies that only K clients will be selected, because only K clients satisfy that $S(D^{k}, D^{i}) \geq \epsilon$. In this way, each client will have the same number of collaborators, which avoids the possible excessive communication overhead.

---

> ### Author Response · Authors · 2022-11-22
> **Need Further Clarification?**
>
> Thanks very much for your constructive comments on our work. We have tried our best to address the concerns. Is there any unclear point so that we should/could further clarify?

---

### Official Review · Reviewer_te42 · 2022-10-26

**Confidence:** 2
**Correctness:** 3
**Technical Novelty And Significance:** 4
**Empirical Novelty And Significance:** 3
**Recommendation:** 6

**Clarity, Quality, Novelty And Reproducibility:**

- Clarity and quality: Writing is clear and easy to follow
- Novelty: The proposed problem setting is realistic and the proposed methods are novel
- Reproducibility: The authors submitted the code to reproduce the result.

**Strength And Weaknesses:**

### Strong points

- This paper tackles a challenging but more realistic federated learning scenario where the shared information may not be consistent across all clients.
- By adopting Normalizing flow based on the bijective model, communicating information about data distributions as a form of data manifold is interesting. Identifying overlaps between the local manifolds and sampling pseudo data with exact likelihood estimation are straightforward approaches to prevent each client from the biased local minima due to the skewed local data.
- The toy experiment demonstrates the effectiveness of the methods intuitively.
- The performance gain is non-trivial on multiple benchmarks.

### Weak points

- The performance gain may be attributed to scaling up the loss by introducing additional cross-entropy loss with the subset of local data. It is necessary to compare the results of the component analysis with each optimal learning rate.
- The paper should evaluate the proposed method on more heterogeneous settings: a large number of clients (lower data points for each client), lower class overlaps, and heterogeneous data split sampled by Dirichlet distribution.

### Questions

- How does the server calculate the overlaps of $U^{\prime i}$ between clients, calculate $U^{\prime i}_s$ while the server does not have any raw data of clients?

**Summary Of The Paper:**

- This paper tackles data heterogeneity of local data distributions in personalized federated learning. Unlike existing works assuming consistent information sharing, this paper introduces fragmented information sharing where the distribution overlaps are not consistent in local clients, which is a more realistic assumption.
- This paper proposes two losses for effective local training: prevents overfitting to local biased data by 1) identifying meaningful overlaps between clients, and 2) generating data from the local manifold with an optimal sampling density.
- Extensive experiments demonstrate the proposed method outperforms existing methods on multiple image classification benchmarks and a real-world clinical data set.

**Summary Of The Review:**

At this point, this paper, in my opinion, is worthy of acceptance.

---

> ### Author Response · Authors · 2022-11-18
> **Response to Reviewer te42**
>
> We would like to thank the reviewer for the positive and very valuable comments. Below are our responses to the comments.
>
> * ****Question 1: It is necessary to compare the results of the component analysis with each optimal learning rate.****
>
>     ****Answer:**** we did the component analysis in our original submission in Sec. C in Appendix.
>
>     We copy the results from our original submission for your easy reference.
>
>  | Dataset | local | $\alpha$ = 0 | $\beta$ = 0 | PCFL |
>  | ---- | ---- | ---- | ---- | ---- |
>  | CIFAR10 | $68.9_{\pm 1.1}$ |  $76.7_{\pm .3}$ |  $79.4_{\pm .7}$ | $84.1_{\pm .8}$ |
>  | FEMNIST | $59.9_{\pm .9}$ |  $75.5_{\pm 1.1}$ |  $82.7_{\pm .8}$ | $89.7_{\pm .2}$ |
>   | CelebA | $69.3_{\pm 1.1}$ |  $82.6_{\pm 2.7}$ |  $80.8_{\pm .9}$ | $85.8_{\pm 1.1}$ |
>    | eICU | $73.7_{\pm .4}$ |  $76.6_{\pm .3}$ |  $77.4_{\pm .5}$ | $78.6_{\pm .4}$ |
>
> We set $\alpha = 0$ to evaluate the performance if we only use the generated data sampled from a learned distribution density, and set $\beta = 0$ to evaluate the performance if we only use the identified overlaps from other clients.
>
> From the above table, we have the following findings:
>
> (1). **every component effectively improves the model learning.** Compared with **local**, both $\alpha = 0$ and $\beta = 0$ achieve a better performance;
>
> (2). **the identified overlaps could achieve more performance gain.** From the above table, $\beta = 0$ performs better, which means that the shared overlaps from other clients facilitate the model learning significantly, compared with the generated data from the learned manifold.
>
> * ****Question 2: The paper should evaluate the proposed method on more heterogeneous settings.****
>
>     ****Answer:**** Following the suggestion, we conduct experiments on FEMNIST on more heterogeneous settings with more clients. We partition the dataset into 400 clients with Dirichlet distribution $Dir_{400}(0.1)$ and $Dir_{400}(0.5)$. We compare the accuracies of our method with the baselines, and the results are shown in the following table.
>
> | method | $Dir_{400}(0.1)$ | $Dir_{400}(0.5)$ |
> | ---- | ---- | ---- |
> | local | $71.8_{\pm .8}$ | $63.6_{\pm .4}$ |
> | FedAvg | $69.1_{\pm .5}$ | $80.6_{\pm 1.7}$ |
> | FedProx | $67.8_{\pm .3}$ | $79.9_{\pm .2}$ |
> | Fed-MTL | $81.8_{\pm .9}$ | $60.1_{\pm .4}$ |
> | PerFedAvg | $82.4_{\pm 1.1}$ | $46.7_{\pm .8}$ |
> | LG-FedAvg | $86.8_{\pm .8}$ | $49.5_{\pm .5}$ |
> | FedPer | $91.1_{\pm .3}$ | $76.8_{\pm .2}$ |
> | FedRep | $91.8_{\pm 1.4}$ | $74.6_{\pm .3}$ |
> | APFL | $79.9_{\pm .9}$ | $60.9_{\pm .5}$ |
> | L2GD | $77.6_{\pm .4}$ | $39.9_{\pm 1.5}$ |
> | Ditto | $91.7_{\pm .7}$ | $82.9_{\pm .4}$ |
> | ****PCFL**** | $96.1_{\pm .4}$ | $88.3_{\pm .6}$ |
>
> With more clients, each client has fewer training samples. Local training shows poor performance ($71.8\%$). Global methods (FedAvg and FedProx) achieve better performance under the less heterogeneous setting ($Dir_{400}(0.5)$), while the performance of personalized methods degrades. Under two settings ($Dir_{400}(0.1)$ and $Dir_{400}(0.5)$), PCFL outperforms all baselines by identifying the informative overlaps for each client.
>
> * ****Question 3: How does the server calculate the overlaps of $U'^{i}$ between clients, calculate $U'^{i}$ while the server does not have any raw data of clients?****
>
>     ****Answer:**** We would like to explain it as follows:
>
>     (1). **$\overline{U'^{i}}$ is identified by the local client.** As you mentioned, the server has no raw data. Only the local client could identify $U'^{i}$ by the local data. The local clients determine the projected manifold $\overline{U'^{i}}$ as the convex hull of $U'^{i}$;
>
>     (2). **the overlaps of $\overline{U'^{i}}$ is calculated by the server.** The local projected manifold $\overline{U'^{i}}$ is a subset of $R^{d'}$, which has no information about the raw data. Similar to the computation of Intersection over Union (IoU), the server could compute the overlapped region using $\overline{U'^{i}}$ without privacy leakage.

---

> ### Author Response · Authors · 2022-11-28
> **Need Further Discussion?**
>
> Dear Reviewer te42:
>
> Many thanks for recognizing the novelty and contributions of our work. In addition, we are also very grateful for the efforts of all reviewers.
>
> As you mentioned, precision collaboration put forwards a new idea for federated learning, which significantly advances the model performance. (The source codes have been made publicly available.)
>
> From the reviews of other reviewers, the main concerns are the expression about motivation and privacy.
>
> We believe our rebuttals could alleviate the concerns raised by the reviewers. Meanwhile, we hope the present scores will not discourage others from discussing our research, and your reply could motivate other reviewers to participate in the discussion.
>
> Best regards,
>
> Authors

---

### Decision · Program_Chairs · 2023-01-20

**Decision:**

Reject

**Justification For Why Not Higher Score:**

There is a mismatch between one of the main claims in the paper, and the theoretical and empirical evidence for it.

As a separate note, the last reviewer raised a concern of anonymity violation, which has not yet been assessed.

**Justification For Why Not Lower Score:**

N/A

**Metareview: Summary, Strengths And Weaknesses:**

The submission proposes a method for personalized federated learning based on manifold estimation.  The claim is that it will better allow clients to make use of overlap between distributions on different clients while achieving privacy preservation by a data abstraction.  A majority of reviewers felt that this submission was below the acceptance threshold.  Main concerns included a lack of theoretical guarantees, both from a statistical generalization perspective and from a privacy perspective.  During the rebuttal process, the reviewers were asked about the main contributions, and the assertion of improved privacy was again raised.  However, privacy should mean privacy by design with provable guarantees, rather than intuitive assertions.  An additional issue was that answers regarding key properties of manifolds were not clear in the discussion, with the authors frequently using the word "agnostic" to describe a manifold in the rebuttal process (while it is used once in the paper).  It is not clear what "mostly agnostic" means, and the responses often lacked formal rigor as a result.  On the balance, the algorithm and empirical results are interesting, but there is a mismatch between the claims of the paper and the theoretical (and to a lesser extent empirical) evidence for them.  Additional theory and experiments backing privacy claims should be given, or these claims should be removed.